# Time series saliency maps: Explaining models across multiple domains

**Christodoulos Kechris** [1]  **Jonathan Dan** [1]  **David Atienza** [1]

## Abstract

Traditional saliency map methods, popularized in computer vision, highlight individual input points that contribute most to a model's output. However, in the context of time series, they offer limited insights because semantically meaningful features are often found in other domains. Thus, we introduce in this paper Cross-domain Integrated Gradients, a generalization of Integrated Gradients that enables feature attributions in any domain formulated as an invertible, differentiable transformation of the time domain. Our derivation extends Integrated Gradients into complex-valued domains, enabling frequency-based attributions, while preserving path independence and completeness. We validate our method via controlled mechanistic experiments, quantitative faithfulness and perturbation-stability tests, and real-world case studies. Across wearable heart-rate extraction, EEG-based seizure detection, and zero-shot forecasting, our proposed Cross-domain Integrated Gradients approach identifies whether predictions rely on heart-rate frequencies or interference, epileptic sources or artifacts, and trend or seasonal components, revealing model behavior that time-domain saliency does not capture. We release an open-source library with TensorFlow, native PyTorch, and Captum support for plug-and-play cross-domain explainability of time-series models.

## 1. Introduction

Saliency maps attribute a model's prediction to individual input features (Selvaraju et al., 2017; Gupta et al., 2022). In domains such as vision and language, these features often align with human-interpretable units (pixels or words), making saliency maps intuitive to inspect (Li et al., 2016).

For time series, this alignment is weaker: neighboring time points do not necessarily correspond to coherent concepts, and predictive factors frequently manifest as *structured latent features* such as frequency components or independent sources (Schröder et al., 2023b;a). As a result, highlighting individual time points can be difficult to interpret and may obscure the mechanisms driving a model's decision.

Signal processing has long addressed this by interpreting signals via structured decompositions: a transform $T$ maps the time series to components $z = T(x)$ whose coordinates correspond to semantically meaningful factors (e.g., sinusoidal frequencies in the Fourier transform (Bracewell, 1989) or statistically independent sources in ICA (Lee, 1998)). The appropriate decomposition is task- and signal-dependent, and choosing $T$ effectively specifies what kinds of features are interpretable.

Schröder et al. demonstrate that time domain saliency can fail when labels depend on latent structure, e.g., frequency content, motivating explanations in an interpretable representation rather than only over time points (Schröder et al., 2023b;a). We build on this insight and treat the explanation domain as a task-dependent design choice: saliency methods should be able to attribute predictions to features in a practitioner-chosen domain, even when the model operates on time points.

To this end, we introduce *Cross-domain Integrated Gradients*, which produces saliency maps directly in a practitioner-chosen explanation domain, e.g., frequency, sources, trend/seasonality, even when the model operates on time samples.

This work builds on a growing line of research that seeks explanations in semantically meaningful representations rather than only over time points. For example, Virtual Inspection Layers (VIL) transport time-domain attributions to the frequency/time-frequency domain through a transform layer using transform-specific propagation rules (Vielhaben et al., 2024). MIX computes Integrated Gradients (Sundararajan et al., 2017) in wavelet view space within a multi-view framework (Tran et al., 2025).

We take the view that the *explanation domain is part of the interpretability task*: choosing a transform $T$ defines the features $z = T(x)$ that are meaningful to inspect for

[1]EPFL, Lausanne, Switzerland. Correspondence to: Christodoulos Kechris <christodoulos.kechris@epfl.ch>.

*Proceedings of the 43$^{rd}$ International Conference on Machine Learning*, Seoul, South Korea. PMLR 306, 2026. Copyright 2026 by the author(s).

a given application. Building on transformed-coordinate attribution, we provide a transform-agnostic toolbox across multiple decompositions and a principled extension of IG to *complex-valued* transform domains (e.g. Fourier, Complex Cepstrum), with axiomatic guarantees enabling faithful attributions in the chosen domain. We validate our method via controlled mechanistic analysis, quantitative faithfulness tests, and real-world case studies, and we release an open-source implementation. Table 1 summarizes the transform-domain coverage of representative time-series explanation methods and situates Cross-Domain IG (CDIG) in this landscape.

| Method | Time | DFT | STFT | DWT | ICA | STL | Cep. |
|---|---|---|---|---|---|---|---|
| Time-IG | ✔ | ✘ | ✘ | ✘ | ✘ | ✘ | ✘ |
| TIMING | ✔ | ✘ | ✘ | ✘ | ✘ | ✘ | ✘ |
| IG/LRP + VIL | ✘ | ✔ | ✔ | ✘ | ✘ | ✘ | ✘ |
| FreqRISE | ✘ | ✔ | ✔ | ✘ | ✘ | ✘ | ✘ |
| FlexTIME | ✘ | ✔ | ✘ | ✘ | ✘ | ✘ | ✘ |
| MIX | ✘ | ✘ | ✘ | ✔ | ✘ | ✘ | ✘ |
| **CDIG (ours)** | ✔ | ✔ | ✔ | ✔ | ✔ | ✔ | ✔ |

*Table 1.* Transform-domain coverage of representative time-series explanation methods. In this work, we instantiate Cross-domain IG in six transform families (DFT, STFT, DWT, ICA, STL, Cepstrum), whereas prior methods typically target specific domains.

In this work, we introduce the following contributions:

- **Cross-domain IG framework**. We operationalize the transformed-coordinate IG as a unified framework for producing saliency maps *in the chosen explanation domain* and instantiate it across six practically relevant transforms (DFT, STFT, DWT, ICA, STL, and Complex Cepstrum).

- **Complex-valued IG with guarantees**. We derive a generalization of the Integrated Gradients for real-valued functions with a complex domain, enabling principled attributions via complex-valued transforms while preserving IG-style axiomatic properties (e.g., completeness).

- **Domain choice is consequential**. We treat the choice of explanation domain as part of the interpretability task. We demonstrate how different domains allow for a better understanding of model behavior on time-series data. We also show that different transforms yield different quantitative behavior.

- **Open-source library**. We provide an open-source library with TensorFlow, native PyTorch and Captum support for cross-domain time series explainability: `https://github.com/esl-epfl/cross-domain-saliency-maps`. The code

for reproducing the results of this paper is available here: `https://github.com/esl-epfl/cross-domain-saliency-maps-paper`.

## 2. Related work

**Time-domain explainability.** Saliency map methods have been applied to time series applications, either by direct application of computer vision-derived methods (Jahmunah et al., 2022; Tao et al., 2024) or by developing dedicated time series saliency approaches (Queen et al., 2023; Liu et al., 2024). Similarly, a time-series adaptation of Integrated Gradients (IG) (Sundararajan et al., 2017) for time-domain attributions was proposed (Jang et al., 2025). To streamline comparisons between time-domain interpretability, an extensive synthetic, multi-channel benchmark was proposed (Ismail et al., 2020). In all cases, these approaches focus on identifying significant regions of the time-domain input that contribute the most to the model's output. Such regions of interest are events that trigger the model's output.

**Cross-domain interpretability.** Time-domain saliency can fail when labels depend on latent structure (Schröder et al., 2023b;a; Theissler et al., 2022; Chung et al., 2024), motivating explanations in semantically meaningful representations. First, representation-restricted post-hoc methods define relevance directly in frequency or time-frequency representations via perturbation or masking (Chung et al., 2024; Brüsch et al., 2025a;b). Second, Virtual Inspection Layers (VIL) (Vielhaben et al., 2024) *transport* time-domain saliency into frequency/time-frequency domains by inserting a transform layer and propagating relevance using transform-specific rules. Third, transformed-coordinate attribution computes IG in a transformed/view space. MIX (Tran et al., 2025) instantiates this idea in a multi-view Haar-DWT setting with segment aggregation and cross-view refinement/selection.

We build on transformed-coordinate attribution to support multiple transforms, including complex-valued domains (e.g., Complex Cepstrum).

**Saliency map evaluation.** Evaluating saliency maps is not a trivial task. A major challenge lies in disentangling saliency map errors from model errors (Kim et al., 2021; Akhavan Rahnama, 2023), complicating validation through comparison with ground truth saliency. The original IG work proposes solving this by relying on a set of desirable axioms, bypassing the necessity of empirical evaluations (Sundararajan et al., 2017). Validation based on insertion / deletion is another approach (Hama et al., 2023; Ismail et al., 2020). These methods empirically evaluate the effect of removing/retaining the most important input features, reinforcing trust in the saliency map method under examination.

# 3. Preliminaries

## 3.1. Problem statement and motivation

We consider a function $f : \mathcal{D}_s \to \mathbb{R}$ representing a deep learning model. The input $\boldsymbol{x} \in \mathcal{D}_s$ is constructed from a continuous-time signal $x(t) \in \mathbb{R}$ after discretizing it at a sampling frequency $f_s$ [Hz] and considering a window of length $L$ seconds: $\boldsymbol{x} = [x_0, ..., x_{n-1}]$, $n = f_s \cdot L$. Now consider a transform $T : \mathcal{D}_S \to \mathcal{D}_T$ that maps the original time domain to a semantically rich explanation target domain $\mathcal{D}_T$. Our task is to construct an informative saliency map that assigns a significance score to each characteristic $z_i = T(\boldsymbol{x})_i$ in the explanation domain. Furthermore, time series transforms can also be complex-valued, e.g. Fourier, our saliency map method should support $z_i \in \mathbb{C}$.

Saliency maps developed in computer vision applications, and in particular IG, provide explanations in the same domain as the model's input, that is, $\mathcal{D}_T = \mathcal{D}_S$. Applying these methods to time-series models results in maps expressed in the time domain.

We summarise the empirical conclusions of relevant works (Schröder et al., 2023b;a; Theissler et al., 2022; Vielhaben et al., 2024), in Proposition 3.1.

**Proposition 3.1.** *The time domain is not always informative in explaining $f$.*

We provide further analytically tractable evidence in support of Proposition 3.1 through our example in Section 3.2 which is in line with the empirical synthetic experiments of (Schröder et al., 2023b). Although this example focuses on the frequency domain, our derivation is transform-agnostic (Section 4) and we expand to more domains in Section 5, providing real-world cases.

## 3.2. Time domain explanation limitations

Consider that the input $\boldsymbol{x}$ is sampled from the signals $x(t) = cos(2\pi\xi t + \phi)$. In this setup, there are two classes of samples depending on the oscillating frequency $\xi$: we set $y = 1$ if $\xi \sim \mathcal{N}(1, 0.5)$ and $y = 2$ if $\xi \sim \mathcal{N}(4, 0.5)$. We design a classifier $f$ to distinguish between these two classes. We opt to manually construct $f$ so that we have full mechanistic understanding of its inner workings. We choose a CNN architecture composed of a single convolutional layer with two channels followed by a ReLU activation and global average pooling $f(\boldsymbol{x}) = AvgPool(ReLU(\boldsymbol{w} * \boldsymbol{x}))$. The kernel of the first channel is a low-pass filter (cutoff at $2.5Hz$), while the second channel kernel is a high-pass filter with the same cutoff (see Figure 1).

Ideally, the model should be fully explained by describing its inner mechanisms. In this particular scenario, we have designed $f$ for this purpose; hence, a formal detailed explanation is available.

*Mechanistic Interpretation* 1. Convolutional channel $i$ allows only frequencies of class $i$ to pass through the output; otherwise, the channel's output is almost zero, not activating. The ReLU and Average Pooling mechanism extract the amplitude of the signal (Kechris et al., 2024a). Hence, the channel $i$ of the model output is only active when samples from class $i$ are processed, leading to the correct classification of the input.

That depth of model understanding is not easily available in larger models, which have been trained on samples. Hence, saliency maps are often used as a proxy. We provide IG explanations of the model $f$ for samples from both classes, expressed in the time and frequency domains (Figure 1). Although time points are periodically highlighted as *more important*, it is not exactly clear how this input influences the model towards producing its output.

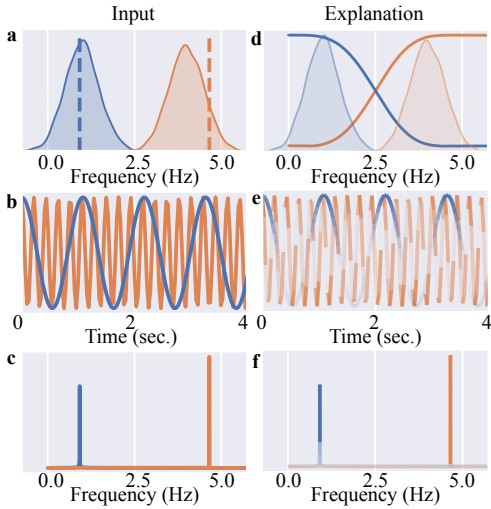

*Figure 1.* **Mechanistic interpretation along with Time and Frequency domain saliency maps. (a)** Distributions of the main frequency, $\xi$, for classes **one** and **two**. We sample one input for each class (vertical dashed lines) for which we generate the saliency maps. **(b)** These two sampled inputs presented in the time and **(c)** frequency domains. **(d)** Illustration of the Mechanistic Interpretation1. We plot the frequency response for the **first** and **second** channels of the CNN. The sample distributions (a) are also overlayed. **(e)** Saliency maps expressed in the time and **(f)** frequency domains.

In contrast, a saliency map expressed in the frequency domain, which we introduce in Section 4, highlights the frequency components that contribute to the final output: for the samples of class one, only the 1 Hz component contributes to the model's output, and accordingly, for class two, the 4 Hz component. Here, this saliency map is much more interpretable. It provides useful information and better aligns with the mechanistic understanding (Mechanistic Interpretation 1) of this model. In Section 4, we show analytically that the frequency-expressed IG, for the data distribution and model of this example, is directly linked to

its mechanistic explanation.

## 3.3. Integrated Gradients

To explain the output of a model $f$ on an input $\boldsymbol{x}$ with a baseline $\hat{\boldsymbol{x}} \in \mathbb{R}^n$, IG generates a saliency map as (Sundararajan et al., 2017):

$$IG_i(\boldsymbol{x}) = (x_i - \hat{x}_i) \int_0^1 \frac{\partial f}{\partial x_i}\bigg|_{\hat{\boldsymbol{x}}+t\cdot(\boldsymbol{x}-\hat{\boldsymbol{x}})} dt \qquad (1)$$

with each element $IG_i(x)$ of the map corresponding to the significance of the input feature $x_i$: saliency is expressed in the same domain as the input. The IG definition relies on two key points from the theory of integrals over differential forms: the line integral definition and Stokes' theorem.

**Line integral definition.** The IG can be derived from the definition of the integral of the differential form $df$ along the line $\boldsymbol{\gamma}(t) = \hat{\boldsymbol{x}} + t(\boldsymbol{x} - \hat{\boldsymbol{x}})$:

$$\int_\gamma df = \int \boldsymbol{\gamma}^* df = \int_0^1 \sum_{i=0}^N \frac{\partial f}{\partial x_i}\gamma_i'(t)dt$$
$$= \sum_{i=0}^N \int_0^1 \frac{\partial f}{\partial x_i}\gamma_i'(t)dt = \sum_{i=0}^N (x_i - \hat{x}_i)\int_0^1 \frac{\partial f}{\partial x_i}dt \tag{2}$$

where $\boldsymbol{\gamma}^* df$ is the pullback of $df$ by $\boldsymbol{\gamma}$: $\boldsymbol{\gamma}^* df = \sum_{i=0}^N \frac{\partial f}{\partial x_i}\gamma_i'(t)dt$ (Do Carmo, 1998). Each individual element of the IG map $IG_i(\boldsymbol{x})$ corresponds to each element of the last sum of eq. 2.

**Stoke's Theorem.** The *Completeness* axiom of the IG (Sundararajan et al., 2017): $f(\boldsymbol{x}) - f(\hat{\boldsymbol{x}}) = \sum IG_i$ is a consequence of the Stokes' Theorem for the case of integral of 1-form: $\int_\gamma df = \int_{\partial\gamma} f = f(\boldsymbol{x}) - f(\hat{\boldsymbol{x}})$, which guarantees path independence: the value of the integral is only dependent on the first and last points of the path, not the path itself.

## 3.4. Saliency maps evaluations

We evaluate Cross-domain IG using complementary evidence: (i) axiomatic guarantees, (ii) mechanistic analysis on a tractable model, (iii) qualitative case studies, and (iv) quantitative faithfulness and perturbation-stability checks.

## 4. Methods

In this section, we define Cross-Domain IG (Section 4.1), and derive it based on the IG principles from Section 3.3. We then analyze it in the complex frequency domain using a simple yet representative convolutional network, highlighting its relation to the network's properties (Section 4.2).

This analysis also provides theoretical grounding for the connection between frequency-domain IG and the Mechanistic Interpretation discussed in Section 3.2. Finally, we detail the implementation of our method.

### 4.1. Cross-domain IG derivation

Let $f : \mathcal{D}_s \to \mathbb{R}$ be a deep neural network operating on a domain $\mathcal{D}_s \subseteq \mathbb{R}^n$. Also, denote $\boldsymbol{x}, \hat{\boldsymbol{x}} \in \mathcal{D}_s$ the input and baseline samples, respectively, as defined by the IG method. We introduce an invertible, differentiable transformation $T : \mathcal{D}_S \to \mathcal{D}_T$ and its inverse $T^{-1}$, which is also differentiable, with $\boldsymbol{z} = T(\boldsymbol{x})$, $\boldsymbol{x} = T^{-1}(\boldsymbol{z})$, and $\mathcal{D}_T \subseteq \mathbb{C}^m$. The cross-domain IG generates the saliency map for $f$, attributing the difference $f(\boldsymbol{x}) - f(\hat{\boldsymbol{x}})$ to the features $\boldsymbol{z}$, expressed in $\mathcal{D}_T$. To define Cross-domain Integrated Gradients, we consider the path integral of model gradients over the transformed feature space:

**Definition 4.1** (Cross-domain Integrated Gradients). Given a model $f : \mathcal{D}_s \to \mathbb{R}$, a transform $T : \mathcal{D}_S \to \mathcal{D}_T$ and its inverse $T^{-1}$, input and baseline samples $\boldsymbol{x}, \hat{\boldsymbol{x}} \in \mathcal{D}_s$ and $\boldsymbol{\gamma}(t)$ the line from $\boldsymbol{z} = T(\boldsymbol{x})$ to $\hat{\boldsymbol{z}} = T(\hat{\boldsymbol{x}})$ the Cross-Domain IG is defined as:

$$IG_i^{\mathcal{D}_T}(\boldsymbol{z}) = 2\int_0^1 \Re\frac{\partial(f \circ T^{-1})}{\partial z_i}\bigg|_{\boldsymbol{\gamma}(t)} \cdot (z_i - \hat{z}_i)dt \tag{3}$$

Note that the original IG, eq. 1, and $IG^{\mathcal{D}_T}$ explain the exact same functionality since $f(\boldsymbol{x})$ and $(f \circ T^{-1})(\boldsymbol{z})$ are equivalent. However, their output saliency maps are expressed in different domains. We now derive Definition 4.1 from the first principles of the original IG method, Section 3.3.

**Derivation sketch.** The original IG is only defined for real inputs. To enable complex-valued transformations, such as the Fourier transform, we extend IG for real-valued functions $g$ with complex inputs $\boldsymbol{z}$, referred to as *Complex IG*. Our derivation builds on the two key points in Section 3.3:

1. **Line integral definition.** We begin by lifting $g : \mathbb{C}^n \to \mathbb{R}$ to a real function $u : \mathbb{R}^{2n} \to \mathbb{R}$ with $u([\boldsymbol{p}, \boldsymbol{q}]) = g(\boldsymbol{p} + j\boldsymbol{q})$, and apply the line-integral argument to $\int_\gamma du$. The end goal is to end up with a sum of integrals $\sum_i \int ...dt$ similar to eq. 2. In the final step, each IG element is defined as the corresponding integral term of the final sum, $\int ...dt$.

2. **Stokes' Theorem.** We define $u$ and derive complex IG to ensure path independence and satisfy the *Completeness axiom*, which may fail for functions of several complex variables (Lebl, 2019). To this end, we first state and prove Lemma 4.2 as an intermediate result. Using Lemma 4.2, we then derive Definition 4.1 using Wirtinger calculus.

**Lemma 4.2.** *Let* $g : \mathbb{C}^n \to \mathbb{R}$, $\boldsymbol{z} = \boldsymbol{p} + j\boldsymbol{q}$, *with* $\boldsymbol{p}, \boldsymbol{q} \in \mathbb{R}^n$, $\boldsymbol{\gamma}(t) = \hat{\boldsymbol{z}} + t(\boldsymbol{z} - \hat{\boldsymbol{z}}), t \in [0, 1]$ *the line from the baseline point* $\hat{\boldsymbol{z}}$ *to the input point* $\boldsymbol{z}$ *and* $\boldsymbol{n}(t) = \Re\boldsymbol{\gamma}(t)$ *and* $\boldsymbol{m}(t) = \Im\boldsymbol{\gamma}(t)$, $\boldsymbol{n}(t), \boldsymbol{m}(t) \in \mathbb{R}^n$. *Then the IG of* $g$ *in* $\boldsymbol{z}$ *is given by:*

$$IG_i^{\mathbb{C}^n}(\boldsymbol{z}) = \int_0^1 \left( \frac{\partial g}{\partial p_i} n_i'(t) + \frac{\partial g}{\partial q_i} m_i'(t) \right) dt \qquad (4)$$

A detailed proof of Lemma 4.2 can be found in Appendix C. From Lemma 4.2, and considering $g(\boldsymbol{z}) = f\left(T^{-1}(\boldsymbol{z})\right)$ and the complex differential form (Range, 1998) $dg = \partial g + \overline{\partial} g$ we can write the complex integrated gradient definition as:

$$IG_i^{\mathbb{C}^n} = 2 \int_0^1 \Re \frac{\partial g}{\partial z_i} \gamma_i'(t) dt \qquad (5)$$

The complete derivation can be found in Appendix D. Notice that Cross-domain IG maintains the *Completeness* property since $\int_\gamma du = u(\boldsymbol{a}(1)) - u(\boldsymbol{a}(0)) = g(\boldsymbol{z}) - g(\hat{\boldsymbol{z}}) = f(\boldsymbol{x}) - f(\hat{\boldsymbol{x}})$, where $u : \mathbb{R}^{2n} \to \mathbb{R}$ s.t. $g(\boldsymbol{p} + j\boldsymbol{q}) = u([\boldsymbol{p}, \boldsymbol{q}])$ and $\boldsymbol{a} = [\boldsymbol{n}, \boldsymbol{m}]$.

*Remark* 4.3. Although definition 4.1 defines a linear path of integration, in our derivation, Eq. 5, the path of integration is a general curve $\boldsymbol{\gamma}(t)$. This enables incorporating into cross-domain IG alternative integration paths/methods to reduce sensitivity to noise (Yang et al., 2023; Kapishnikov et al., 2021).

**Cross-Domain IG for real-valued inputs.** If $g$ processes real-valued inputs, then eq. 5 is equivalent to eq. 1: since $g(\boldsymbol{z}) = g(\boldsymbol{p} + j0)$, $\partial g / \partial q = 0$, $\partial g / \partial z = (1/2)\partial g / \partial p$. Thus, if $\mathcal{D}_T \subseteq \mathbb{R}^n$ the cross-domain IG can equivalently be expressed as $IG_i^{\mathcal{D}_T}(\boldsymbol{z}) = (z_i - \hat{z}_i) \int \frac{\partial(f \circ T^{-1})}{\partial z_i} dt$.

*Remark* 4.4. For $T$ chosen as a Haar-DWT view transform, this recovers the view-space IG used in MIX (Tran et al., 2025)

*Remark* 4.5. In IG (Sundararajan et al., 2017), the baseline $\hat{\boldsymbol{x}}$ is defined as the point without information about the original model inference. For Cross-domain IG, if such a signal-domain baseline exists and $T$ is invertible, then the corresponding explanation-domain baseline is immediately defined $\hat{\boldsymbol{z}} = T(\hat{\boldsymbol{x}})$. Conversely, when the intended reference is more naturally specified in the explanation domain, one may define $\hat{z} \in \mathcal{D}_T$ directly and use $\hat{\boldsymbol{x}} = T^{-1}(\hat{\boldsymbol{z}})$, provided that $\hat{\boldsymbol{x}} \in \mathcal{D}_S$. Thus, the same semantic reference can be expressed in whichever domain is more natural for the application. In all cases, the baseline should be chosen according to the interpretability question: for example, a zero-signal baseline asks which components contribute relative to absence of signal, while other baselines define different contrastive explanations.

## 4.2. Complex IG on a simple model

Prior work analytically studies a minimal single-layer convolutional network, demonstrating that IG can collapse into an *edge detector*, producing misleading saliency maps (Adebayo et al., 2018). Although this exposes a failure mode of the IG in the input domain, we show that Complex-IG faithfully reflects the inner mechanisms of a simple convolutional network in the frequency domain. In direct parallel, we derive a closed-form link between the complex IG saliency map of a CNN and the frequency response of its filters. Building on the example in Section 3.2, we work on a simple CNN and prove that Complex-IG highlights each filter's gain at its corresponding input frequency.

Let $f$ be a convolutional neural network composed of a single convolutional layer (1 channel) followed by a ReLU operation and Global Average Pooling: $f(\boldsymbol{x}) = AvgPool(ReLU(\boldsymbol{w} * \boldsymbol{x}))$. We begin with the case in which $f$ processes windows sampled from single-component sinusoidal signals $x(t) = a_i \cdot cos(2\pi\xi_i t + \phi)$, $a_i > 0$. Then, the output $f(\boldsymbol{x})$ is (Kechris et al., 2024a): $f(\boldsymbol{x}) = \frac{a_i b_i}{\pi}$, with $b_i$ the amplification of the filter $\boldsymbol{w}$ at frequency $\xi_i Hz$: $b_i = \|\sum_n w_n e^{-j2\pi\xi_i n}\|$. We employ the Complex IG method on $f$ with baseline input $\hat{\boldsymbol{x}} = \boldsymbol{0}$, $f(\boldsymbol{0}) = 0$. This yields $IG_k^{\mathbb{C}^n} = 0$, $\forall k \neq i$ and $\sum_i IG_i^{\mathbb{C}^n} = f(\boldsymbol{x}) - f(\hat{\boldsymbol{x}})$. Thus,

$$IG_i^{\mathbb{C}^n} = f(\boldsymbol{x}) = \frac{a_i b_i}{\pi} \qquad (6)$$

This links $IG_i^{\mathbb{C}^n}$ to the output frequency content $a_i b_i$ and, by extension, to the convolutional filter's frequency response. An example for the model of Section 3.2 is presented in Figure 5 (Appendix E).

## 4.3. Implementation

Autograd (pytorch / tensorflow) allows for automatic differentiation with complex variables using Wirtinger calculus (Kreutz-Delgado, 2009). Thus, the complex IG can be directly approximated by autograd, using Definition 4.1 or Lemma 4.2, with the detail that Autograd (in both libraries) calculates the conjugate of the complex partial derivative. For the integral calculation, we use a summation approximation similar to (Sundararajan et al., 2017). The algorithms for estimating cross-domain IG using Lemma 4.2 or Definition 4.1 are presented in Algorithms 1 and 2 in the appendix.

*Remark* 4.6. The numerical approximation of the integral in Definition 4.1 follows the same repeated-gradient pattern as standard IG: the dominant cost in both methods is evaluating gradients along the integration path. Cross-domain IG adds a transform-dependent cost because gradients must also be propagated through $T^{-1}$ (see line 9 in Algorithm 2 in the Appendix). For fixed transforms such as Fourier transforms, this overhead is modest relative to the repeated model forward/backward passes. For transforms such as ICA, there may also be an upfront cost to construct the representation, but this is not specific to CDIG and would be required by any explainer operating in the ICA domain. We quantify

this overhead in Appendix B, where Fourier-domain CDIG adds between 27.68% overhead for a small shallow CNN and 1.77% for the largest tested configuration.

# 5. Experiments

## 5.1. Qualitative evaluation

We deploy cross-domain IG in a range of time-series applications and models spanning the three main time-series task types: regression (section 5.1.1), classification (section 5.1.2), and forecasting (Section 5.1.3). In all cases, the models operate on time-domain inputs. For each application, we fix the explanation domain before inspecting the resulting saliency maps, following the interpretability question and signal semantics: we (i) characterise the signal from a signal-processing perspective, (ii) state interpretability task: *what do we want to learn about our model's behavior through a saliency map?*, (iii) select an explanation domain using domain knowledge, and (iv) summarise actionable insights from the resulting attributions. Time-Domain IG attributions and additional qualitative examples are provided in Appendix I and J. All IG and CDIG attributions in Section 5 use the zero-signal baseline.

### 5.1.1. HEART RATE EXTRACTION FROM PHYSIOLOGICAL SIGNALS

We use the KID-PPG (Kechris et al., 2024b), a deep convolutional model with attention, to extract heart rate (HR) from photoplethysmography (PPG) signals collected from a wrist-worn wearable device. We use signals from the PPGDalia dataset (Reiss et al., 2019). For a time window small enough for the HR frequency, $\xi_{hr}$, to be considered constant, a clean PPG signal can be modeled as (Kechris et al., 2024b):$x(t) = a_1 cos(2\pi \cdot \xi_{hr} \cdot t + \phi) + a_2 cos(2\pi \cdot (2\xi_{hr}) \cdot t + \phi)$, with $a_1 > a_2$. However, external signals are also usually present in PPG recordings (Reiss et al., 2019; Kechris et al., 2024b). These *interferences* are not created by the heart and are preventing the model from making accurate HR inferences.

*Remark* 5.1. KID-PPG processes PPG signals that contain both heart-related components and external interference. A trustworthy model should *base* the inferred heart rate on heart-related signals only, filtering out all other sources of noise.

**Interpretability task.** Given a PPG sample and KID-PPG's HR inference, determine whether the model is focusing on heart-related information or external interference.

**Problem-specific transformation.** Since our understanding of this application is mostly frequency-based, we have selected the frequency domain, using the Fourier transform, as the explanation target domain. Hence, the frequency-domain IG highlights individual frequencies as being important to the final model inference. This allows us to investi-

gate whether the HR inference is produced by components related to the heart or by external interference.

An illustration of two PPG inputs and the corresponding frequency-domain IGs is presented in Figure 2. The frequency IG identifies samples in which the model infers heart rate from external interference, thus limiting the reliability of the model's output.

*Remark* 5.2. Frequency-domain IG highlights whether KID-PPG inference is **trustworthy** (based on heart oscillations) or **spurious** (based on motion-induced artifacts).

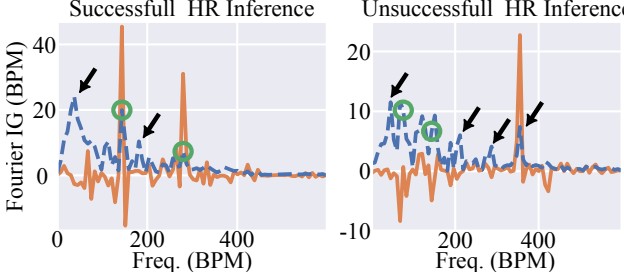

*Figure 2.* **Frequency-domain IG on heart rate inference model.** The PPG signal includes components from the heart rate and other components attributed to external interference (denoted with arrows), e.g. motion. **Left:** Sample with a small inference error 0.93 beats-per-minute (BPM). The IG highlights the two heart components located at $hr$ and $2 \cdot hr$ (second harmonic), with more weight given to the actual heart rate frequency. **Right:** PPG sample with high inference error (26.78 BPM). IG coefficients highlight frequency components which are not related to the heart.

### 5.1.2. ELECTROENCEPHALOGRAPHY-BASED EPILEPTIC SEIZURE DETECTION

We use the `zhu-transformer` (Zhu and Wang, 2023), which performs seizure detection on scalp-electroencephalography (EEG). We analyze a recording from the Physionet Siena Scalp EEG Database v1.0.0 (Detti, 2020; Detti et al., 2020; Goldberger et al., 2000). In EEG a single channel captures the electrical activity of multiple *sources*: e.g., epileptic activity, muscle interference, or electrical noise.

*Remark* 5.3. A seizure classification model processes the aggregated activity of all sources in the EEG. The model should isolate only the epileptic activity, filtering out all others, to reach a trustworthy inference.

**Interpretability task.** Given an EEG recording and the corresponding `zhu-transformer` seizure classification, we want to identify the *sources* on which the model based its inference.

**Problem-specific transformation.** We chose Independent Component Analysis (Lee, 1998) (ICA) as our transform of choice. ICA isolates the activity of each individual source to a source-specific channel (Independent Component), as-

suming statistical independence between the sources. This allows the ICA-domain IG to produce attributions for each individual isolated source, thereby providing insights into our interpretability task (Figure 3).

*Remark* 5.4. ICA-IG highlights whether `zhu-transformer` inference is based on known components of epileptic seizure activity or other components irrelevant to the seizure, thus further reinforcing **trust** in the model decision.

### 5.1.3. FOUNDATION MODEL TIME SERIES FORECASTING

We use TimesFM (Das et al., 2024) time-series foundation model to explain forecasting outputs. We perform zero-shot forecasting, without any fine-tuning, on a time series with exponential trend and seasonal components (Figure 4).

*Remark* 5.5. A time-series forecasting model should be equally successful in modeling both the trend and the season to achieve a low-error, long-horizon forecast.

**Interpretability task.** Given a time-series input and the TimesFM forecast, determine whether the trend or the season is more difficult to model in the long-horizon forecast setting.

**Task-specific transform.** To isolate the relevant *concepts*, we chose Seasonal-Trend decomposition using LOESS (STL) (Cleveland et al., 1990) to decompose the input time series into trend and seasonal components.

This attribution domain allows us to study the model's behavior for long-term forecasting horizons where the forecast error increases: the model underestimates the overall trend, while the estimation of the seasonal component presents a smaller error.

*Remark* 5.6. Seasonal-Trend IG reveals that TimesFM underweights the trend, degrading long-horizon forecasts. This offers concrete insights to improve model behavior.

### 5.2. Quantitative evaluation

We use three complementary quantitative checks. First, feature-level intervention tests ask whether highly attributed components have a large effect on the model output. Second, benchmark comparisons ask how CDIG performs relative to existing time-series explanation methods. Third, perturbation-stability tests ask whether the attribution structure changes smoothly under additive input noise. Together, these evaluations assess faithfulness, comparative performance, and robustness.

### 5.2.1. FAITHFULNESS ON THE REAL-WORLD USE CASES

We evaluate faithfulness via feature-level insertion/deletion on the two real-world models from Sections 5.1.1, 5.1.2 (protocol and results in Appendix H). For PPG, deleting

the top 3% Fourier features ranked by Cross-domain IG changes the predicted heart-rate output by 66.39 BPM on average, compared to 10.13 BPM when deleting the top 3% time-domain features. Conversely, inserting the top 3% of frequency features yields a smaller deviation from the original HR inference (37.98 BPM) than inserting the top 3% time-domain features (94.58 BPM).

For EEG, we delete/retain a single IC component. Retaining the most important IC preserves the model output substantially better than a random IC: the average change in seizure probability is 0.0696 vs 0.4396 for randomly retained IC (smaller is better). In the deletion test, deleting the most important IC causes a much larger change (0.177) than deleting a random IC (0.0083), as expected (larger is better).

### 5.2.2. COMPARISONS ACROSS METHODS AND DOMAINS

We quantify explanation faithfulness using Cumulative Prediction Difference (CPD) from TIMING (Jang et al., 2025) and compare against TIMEX++ (Liu et al., 2024) and TIMING (Jang et al., 2025) in Table 2. Following the TIMING protocol, we report CPD under two substitution average and zero as defined in (Jang et al., 2025).

Overall, Cross-domain IG is consistently stronger than TIMEX++ and competitive with TIMING. Importantly, different domain instantiations of our framework excel on different datasets (e.g., Cepstrum on PAM/Epilepsy/Freezer, Fourier on Wafer), quantitatively supporting that the explanation domain is task-dependent. The "Avg." and "Zero" columns refer to the CPD substitution protocol of Jang et al. (Jang et al., 2025), not to the IG baseline, which remains the zero-signal baseline as specified in Section 5.1. To assess sensitivity to this baseline choice, we repeat the same benchmark with a dataset-mean IG baseline in Appendix K.

Where direct frequency-domain baselines are available, AudioMNIST (Becker et al., 2024), Cross-domain IG achieves comparable faithfulness and complexity to FreqRISE (Brüsch et al., 2025a) and VIL (Vielhaben et al., 2024) (Appendix G). Cross-domain IG directly supports expanding to new transforms for which we provide additional evaluations for Discrete Wavelet Transform (DWT) and Complex Cepstrum. In the same benchmark, comparing Cross-domain IG across domains shows that Cepstrum yields the best faithfulness for digit classification, while Time-Frequency yields the best faithfulness for gender classification, further reinforcing the task-dependence of the explanation domain. DWT presented the best complexity for both tasks.

We note that these benchmarks measure faithfulness under standardized protocols. They do not capture the semantic utility of a domain choice, which we illustrate in the case studies (Section 5.1).

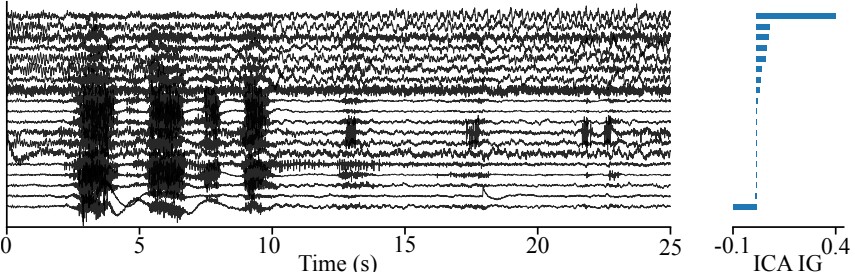

*Figure 3.* **ICA-domain IG on seizure detection model.** The ICA components are sorted from the component with the highest IG significance (top) to the lowest (bottom). **Left:** 19 output channels calculated from ICA on the original EEG channels. The first channel contains the majority of the epileptic activity, which is visible as an evolving pattern of spike-and-wave discharges at $\sim 4.5$ Hz. Some epileptic activity can also be found in the second channel. Significant muscle artifacts are isolated in the 9th-19th channels between 4 and 10 seconds. **Right:** IG saliency map calculated on the channel components. The map identifies the first channel as the most significant channel in detecting this sample as epileptic. Some significance, although much less, is also given to the next four channels. The channels corresponding to interference components do not get any significance in the output of the classifier. The last channel *tends to tilt* the classifier towards a non-epileptic output.

| | PAM | | Boiler | | Epilepsy | | Wafer | | Freezer | |
| --- | --- | --- | --- | --- | --- | --- | --- | --- | --- | --- |
| | Avg. | Zero | Avg. | Zero | Avg. | Zero | Avg. | Zero | Avg. | Zero |
| TIMEX++ | $0.057 \pm 0.004$ | $0.070 \pm 0.004$ | $0.124 \pm 0.028$ | $0.208 \pm 0.043$ | $0.030 \pm 0.004$ | $0.032 \pm 0.004$ | $0.000 \pm 0.000$ | $0.000 \pm 0.000$ | $0.216 \pm 0.056$ | $0.216 \pm 0.056$ |
| TIMING | $0.463 \pm 0.007$ | $0.602 \pm 0.033$ | **$1.259 \pm 0.065$** | $1.578 \pm 0.085$ | $0.057 \pm 0.005$ | $0.060 \pm 0.005$ | $0.674 \pm 0.014$ | $0.674 \pm 0.014$ | $0.409 \pm 0.062$ | $0.409 \pm 0.109$ |
| Fourier (Ours) | $0.207 \pm 0.008$ | $0.624 \pm 0.035$ | $1.233 \pm 0.038$ | $1.335 \pm 0.047$ | $0.141 \pm 0.03$ | $0.144 \pm 0.004$ | **$0.772 \pm 0.036$** | **$0.772 \pm 0.036$** | $0.252 \pm 0.062$ | $0.252 \pm 0.062$ |
| Cepstrum (Ours) | **$1.183 \pm 0.040$** | **$1.230 \pm 0.044$** | $1.246 \pm 0.057$ | **$1.665 \pm 0.080$** | **$0.795 \pm 0.041$** | **$0.712 \pm 0.028$** | $0.244 \pm 0.021$ | $0.647 \pm 0.023$ | **$0.646 \pm 0.192$** | **$0.726 \pm 0.165$** |

*Table 2.* Performance comparison of Cross-domain IG with TIMEX++ (Liu et al., 2024) and TIMING (Jang et al., 2025). We evaluate Cumulative Prediction Difference (CPD) aggregated over 5 random repetitions (average $\pm$ standard error). Higher is better.

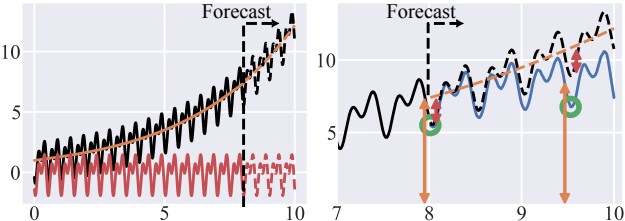

*Figure 4.* **Seasonal-Trend IG on time series foundation model. Left: Input** time series decomposed via STL into **trend** and **seasonality**. **Right:** Zero-shot **forecasting** using TimesFM with **Seasonal**-**Trend** IG. For a small horizon, one step ahead prediction (**first circle**), TimesFM forecasts accurately. Of output, 7.5 units are attributed to trend (⇕), aligning with ground truth (dashed orange) and similarly $-1.96$ units to seasonality (⇕). For a longer horizon (**second circle**) the forecast absolute error rises from 0.2 to 2.14. Most of it stems from the model's underestimation of the **trend** (21% relative error), while the **seasonal** effect is correctly captured by the model (5.1% relative error).

### 5.2.3. ROBUSTNESS TO INPUT PERTURBATIONS

We additionally evaluate attribution stability under additive input noise in Appendix L. In the controlled Fourier setting of Section 4.2 , frequency-domain CDIG retains most attribution mass in the target frequency support at high and moderate SNRs, with degradation occurring gradually as noise becomes dominant. On the real-world PPG task from Section 5.1.1 , Fourier-domain CDIG is more stable than time-domain IG when comparing clean and noisy in-

puts whose predictions remain close to the clean prediction. These perturbation experiments provide a complementary stability check: rather than measuring faithfulness through feature removal or CPD, they test whether the attribution structure changes smoothly under input noise.

## 6. Discussion

Across the qualitative case studies (Section 5.1), Cross-Domain IG produces attributions in domains that align with practitioner reasoning (frequency for PPG, ICA for EEG, trend/seasonality for forecasting). Quantitatively, insertion /deletion benchmarks (Section 5.2) indicate that cross-domain attributions can be more faithful than time-domain attributions and remain competitive with strong time-series saliency methods. We demonstrate comparable faithfulness/complexity metrics to existing frequency-domain saliency methods (Appendix G), while also demonstrating that our method applies to a broader class of transforms, including complex-valued ones.

**Specifying the explanation domain.** The preceding case studies illustrate that our method does not select the explanation domain automatically, rather the transform is part of the interpretability specification. Its role is to define the coordinates over which the model output is decomposed. In practice, the domain should be chosen from the question being asked about the signal, before computing or inspecting

saliency maps. For example, Fourier coordinates are appropriate when the question concerns oscillatory or spectral structure, ICA coordinates when it concerns source-level activity, and STL coordinates when it concerns trend and seasonality. There is therefore no universally best transform for all time-series problems. Cross-domain IG is instantiated in one chosen explanation domain at a time; it is not a multi-view aggregation method, although several pre-specified domains can be analyzed separately as complementary views. When several domains are scientifically plausible, they should be treated as complementary pre-specified views, or assessed through faithfulness checks and domain knowledge. They should not be selected post hoc because one domain yields a cleaner or more expected visualization. Such *"domain-shopping"* can create misleading confidence even when the attribution computation itself is faithful. Thus, cross-domain IG provides principled attributions once the explanation domain and baseline are specified, but it does not remove the need for domain expertise in making that specification.

**Limitations.** While Cross-Domain IG addresses the misalignment between time-domain saliency maps and latent structure, it inherits several generic limitations of IG. We use a straight-line integration path and a single zero baseline; alternative paths or baselines can change the quantitative attributions, and existing variants of IG that stabilize these choices are directly applicable but not explored here. Our framework assumes an invertible, differentiable transform and is instantiated only with transforms for which this assumption is reasonable, e.g., Fourier, ICA, and seasonal-trend decomposition. Non-invertible or approximately invertible representations are outside our guarantees. Finally, as discussed above, the method presupposes that the practitioner can choose a meaningful explanation domain. If this choice is poor, based on incorrect prior knowledge, or is made post-hoc, Cross Domain IG can yield clean-looking but semantically misleading maps (see Appendix O for a detailed discussion). A further limitation is that our qualitative and quantitative evaluations serve different purposes. In the qualitative case studies, we intentionally leverage domain knowledge to choose an explanation space that matches the interpretability question. In contrast, benchmark comparisons necessarily adopt fixed protocols and standardized transform choices to enable reproducible scoring across methods and datasets. This gap reflects an open challenge in time-series explainability: how to evaluate *domain appropriateness* and practitioner utility, not only faithfulness under a single masking protocol. Developing benchmark suites and metrics that account for task-dependent domain selection is an important direction for future work.

## 7. Conclusions

We introduce a novel generalization of the Integrated Gradients method, which enables saliency map generation in any invertible, differentiable transform domain, including complex spaces. As transforms capture high-level interactions between input points, our methods enhance model explainability, especially in time-series data where individual time-point features are often uninformative. We demonstrated the versatility of Cross-Domain Integrated Gradients, applying it to a diverse set of time-series tasks, model architectures, and explanation target domains. Fields where time signals are extensively used, such as healthcare, finance, and environmental monitoring, could benefit from domain-specific saliency maps. In particular, with the recent rise of time-series foundation models, our method provides a powerful investigative tool for examining model behavior. We release an open-source library to enable broader adoption of cross-domain time-series explainability.

## Acknowledgements

We thank Nikolaos Tsakanikas for insightful feedback on the methodological formulation and derivation. This research was partially supported by IMEC through a joint PhD grant for ESL-EPFL. Also, this work was supported in part by the Swiss NSF, grant no. 10.002.812, titled "Edge-Companions: Hardware/Software Co-Optimization Toward Energy-Minimal Health Monitoring at the Edge".

## Impact statement

This work enables time-series model interpretability by generating saliency maps in meaningful domains, such as frequency or independent component bases. Fields where time signals are extensively used, such as healthcare, finance and environmental monitoring, could benefit from domain-specific saliency maps. In particular, with the recent rise of time-series foundation models, our method provides a strong investigation tool for inspecting model behavior.

Risks may arise if the selected explanation target domain is inappropriate, if domains are selected post hoc based on visually convincing saliency maps, or if saliency maps are over-interpreted as causal evidence. It is important to note that the saliency map provides only feature significance scores. Interpreting these scores requires domain expertise. We encourage a holistic interpretative approach to integrating domain knowledge with cross-domain saliency maps. We also caution that this method alone cannot function as definitive proof of the behavior of the model. Responsible usage of the method should take into consideration model, data, and transformation limitations, especially in high-stakes settings such as in healthcare. We elaborate on the limitations of our method in Appendix O.

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

## A. Cross-domain IG Algorithms

---

**Algorithm 1** Complex Target Domain IG

---

**Input:** $f(\cdot), x, \hat{x}, n_{iter}$
**Output:** $IG$
1: $i \leftarrow 1$
2: $sum\_real \leftarrow 0$
3: $sum\_imag \leftarrow 0$
4: $tape\_real \leftarrow tensorflow.GradientTape()$
5: $tape\_imag \leftarrow tensorflow.GradientTape()$
6: $\hat{z} \leftarrow T(\hat{x})$
7: **for** $i \leq n_{iter}$ **do**
8: $\quad X \leftarrow T(x)$
9: $\quad z \leftarrow \hat{z} + (z - \hat{z}) \cdot i/n_{iter}$
10: $\quad re\_z \leftarrow \Re z$
11: $\quad im\_z \leftarrow \Im z$
12: $\quad tape\_real.watch(\text{re\_z})$
13: $\quad tape\_imag.watch(\text{im\_z})$
14: $\quad \hat{z} \leftarrow re\_z + j \cdot im\_z$
15: $\quad x_{rec} \leftarrow T^{-1}(\hat{z})$
16: $\quad y \leftarrow f(x_{rec})$
17: $\quad re\_dy \leftarrow tape\_real.gradient(y, re\_z)$ {Calculate $\frac{\partial g}{\partial p_i}$}
18: $\quad im\_dy \leftarrow tape\_imag.gradient(y, im\_z)$ {Calculate $\frac{\partial g}{\partial q_i}$}
19: $\quad sum\_real \leftarrow sum\_real + re\_dy$
20: $\quad sum\_imag \leftarrow sum\_imag + im\_dy$
21: $\quad i \leftarrow i + 1$
22: **end for**
23: $sum\_real \leftarrow sum\_real/n_{iter}$
24: $sum\_imag \leftarrow sum\_imag/n_{iter}$
25: $IG = \Re z - \hat{z} \cdot sum\_real + \Im z - \hat{z} \cdot sum\_imag$

---

## B. Runtime overhead

We compare standard IG with Fourier-domain CDIG on CNNs with fixed width of 64 channels. These experiments isolate the additional cost of the transform/inverse-transform step after the explanation domain has been specified. For each configuration, we report the percentage runtime overhead of CDIG relative to standard IG. CDIG adds a measurable but modest overhead. This overhead is largest for small models and short inputs, where the transform-related cost is a larger fraction of the total attribution time. As model depth or input length increases, the repeated forward/backward passes through the network dominate, and the relative overhead decreases.

These results are consistent with the implementation discussion in Section 4.3: CDIG is computationally close to IG, with the CDIG-specific overhead coming from differentiation through the fixed transform/inverse-transform layer. For transforms such as ICA, an additional cost may be re-

---

**Algorithm 2** Complex Target Domain IG with complex differential

---

**Input:** $f(\cdot), x, \hat{x}, n_{iter}$
**Output:** $IG$
1: $i \leftarrow 1$
2: $sum \leftarrow 0$
3: $tape \leftarrow tensorflow.GradientTape()$
4: $\hat{z} \leftarrow T(\hat{x})$
5: **for** $i \leq n_{iter}$ **do**
6: $\quad z \leftarrow T(z)$
7: $\quad z \leftarrow \hat{z} + (z - \hat{z}) \cdot i/n_{iter}$
8: $\quad tape.watch(z)$
9: $\quad x_{rec} \leftarrow T^{-1}(z)$
10: $\quad y \leftarrow f(x_{rec})$
11: $\quad dy \leftarrow tape.gradient(y, X)$
12: $\quad sum \leftarrow sum + \overline{dy}$
13: $\quad i \leftarrow i + 1$
14: **end for**
15: $sum \leftarrow sum/n_{iter}$
16: $IG = 2\Re(z - \hat{z}) \cdot sum$

---

| Fixed input size: 256 | | | | |
|---|---|---|---|---|
| Depth | 3 | 6 | 9 | 12 |
| Overhead (%) | 27.68 | 12.09 | 8.93 | 10.10 |
| **Fixed depth: 2 layers** | | | | |
| Input length | 256 | 512 | 1024 | 2048 | 4096 |
| Overhead (%) | 23.65 | 24.99 | 21.29 | 15.14 | 11.54 |
| **Fixed depth: 9 layers** | | | | |
| Input length | 256 | 512 | 1024 | 2048 | 4096 |
| Overhead (%) | 10.98 | 8.63 | 6.21 | 3.42 | 1.77 |

*Table 3.* Runtime overhead of Fourier-domain CDIG relative to standard IG on CNNs with fixed width of 64 channels. The reported values isolate the additional transform/inverse-transform cost once the explanation domain has been specified.

quired to construct the representation itself, but this is not specific to CDIG, it would also be required by any attribution method operating in that representation.

## C. Proof of Lemma 4.2

*Lemma.* Let $g : \mathbb{C}^n \to \mathbb{R}$, $\boldsymbol{z} = \boldsymbol{p} + j\boldsymbol{q}$, with $\boldsymbol{p}, \boldsymbol{q} \in \mathbb{R}^N$, $\boldsymbol{\gamma}(t) = \hat{\boldsymbol{z}} + t(\boldsymbol{z} - \hat{\boldsymbol{z}}), t \in [0, 1]$ the line from the baseline point $\hat{\boldsymbol{z}}$ to the input point $\boldsymbol{z}$ and $\boldsymbol{n}(t) = \Re\boldsymbol{\gamma}(t)$ and $\boldsymbol{m}(t) = \Im\boldsymbol{\gamma}(t), \boldsymbol{n}(t), \boldsymbol{m}(t) \in \mathbb{R}^n$. Then the IG of $g$ in $\boldsymbol{z}$ is given by:

$$IG_i^{\mathbb{C}^n}(\boldsymbol{z}) = \int_0^1 \left( \frac{\partial g}{\partial p_i} n_i'(t) + \frac{\partial g}{\partial q_i} m_i'(t) \right) dt \quad (7)$$

*Proof.* Let $u : \mathbb{R}^{2n} \to \mathbb{R}$ such that $g(\boldsymbol{z}) = u(\boldsymbol{w}), \forall \boldsymbol{z} =$

$p + jq, w = [p, q]$. For the differential form of $u$:

$$du := \sum_{i=0}^{2N} \frac{\partial u}{\partial w_i} dw_i \qquad (8)$$

Similarly to the $g(z)$–$u(w)$ equivalence, we consider the equivalence between $\gamma(t)$ and $a(t) = [n(t), m(t)] \in \mathbb{R}^{2n}$. Then the pullback of $du$ by $a$ is :

$$a^* du := \sum_{i=0}^{2N} \frac{\partial u}{\partial w_i} a_i'(t) dt \qquad (9)$$

Denoting with $a_i'$ the i-th element of $da/dt$. The line integral of $u$ along the line defined by $a$ is:

$$\int_\gamma du = \int_\gamma a^* du = \int_0^1 \sum_{i=0}^{2N} \frac{\partial u}{\partial w_i} a_i'(t) dt$$
$$= \sum_{i=0}^{2N} \int_0^1 \frac{\partial u}{\partial w_i} a_i'(t) dt$$

Due to the equivalence between $w$ and $p, q$, and $u$ and $g$, the latter sum can be formulated as :

$$\int_\gamma du = \sum_{i=0}^{N} \left( \int_0^1 \frac{\partial g}{\partial p_i} n_i'(t) dt + \int_0^1 \frac{\partial g}{\partial q_i} m_i'(t) dt \right) \quad (10)$$
$$= \sum_{i=0}^{N} \int_0^1 \left( \frac{\partial g}{\partial p_i} n_i'(t) + \frac{\partial g}{\partial q_i} m_i'(t) \right) dt \qquad (11)$$

, which concludes the derivation. $\square$

## D. Derivation of Definition 4.1

From Lemma 4.2, we conclude with Definition 4.1 by considering $g(z) = f\left(T^{-1}(z)\right)$ and the complex differential form (Range, 1998):

$$dg = \partial g + \bar{\partial} g \qquad (12)$$

with $\partial g = \sum \partial g / \partial z_i dz_i$, $\bar{\partial} g = \sum \partial f / \partial \overline{z_i} \overline{dz_i}$. The complex partial derivatives are defined as (Range, 1998) $\partial / \partial z_i = 1/2(\partial/\partial p - j\partial/\partial q)$ and $\partial/\partial \overline{z_i} = 1/2(\partial/\partial p + j\partial/\partial q)$. Then the pullback of $dg$ by $\gamma$ is :

$$\gamma^* dg = \sum \frac{\partial g}{\partial z_i} \gamma_i'(t) dt + \sum \frac{\partial g}{\partial \overline{z_i}} \overline{\gamma_i'(t)} dt \qquad (13)$$

Since $g \in \mathbb{R}$, $\partial g / \partial \overline{z} = \overline{(\partial g / \partial z)}$; thus:

$$\gamma^* dg = 2\Re \sum \frac{\partial g}{\partial z_i} \gamma_i'(t) dt \qquad (14)$$

Expanding the product into its real and imaginary parts produces the same form as eq. 10:

$$\gamma^* dg = 2\Re \sum \frac{1}{2} \left( \frac{\partial g}{\partial p_i} - j \frac{\partial g}{\partial q_i} \right) (n_i' + j m_i'(t)) dt$$
$$= \sum \left( \frac{\partial g}{\partial p_i} n_i'(t) + \frac{\partial g}{\partial q_i} m_i'(t) \right)$$

Therefore, the complex integrated gradient definition can be rewritten as:

$$IG_i^{\mathbb{C}^n} = 2 \int_0^1 \Re \frac{\partial g}{\partial z_i} \gamma_i'(t) dt \qquad (15)$$

## E. Relationship between frequency-domain IG and frequency response

We probe the two convolutional channels of section 3.2 with sinusoidal signals at varying frequencies, $\xi_i$:

$$x_i(t) = cos(2\pi \xi_i t + \phi) \qquad (16)$$

For each input, we perform frequency-domain IG, which yields a saliency map described by eq. 6. We aggregate all produced IGs and compare them to each filter's frequency response:

$$b_i = \| \sum_n w_n e^{-2\pi \xi_i n} \| \qquad (17)$$

The results are presented in Figure 5.

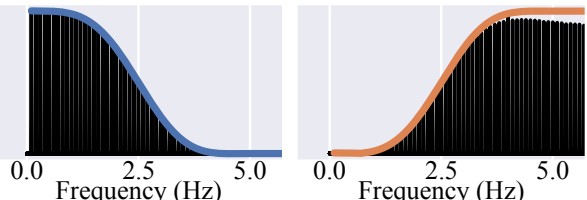

*Figure 5.* Frequency response (blue - orange) and frequency integrated gradients (black) for the two channels of the model of Section 3.2. We probe the model, performing frequency IG on samples with varying base frequencies.

## F. Relation to Virtual Inspection Layers

We demonstrate here the equivalence between Eq. 5 and the Virtual Inspection Layer (Vielhaben et al., 2024) for the case of the Discrete Fourier Transform (DFT) domain saliency maps.

Denote the DFT transform $z = Tx$ with :

$$T_{nk}^{-1} = \frac{1}{\sqrt{N}} e^{2\pi kn/N} \qquad (18)$$

Thus, from eq.5

$$IG_k^{DFT} = 2 \int_0^1 \Re \sum_{n=0}^{N-1} \frac{\partial f}{\partial x_n} T_{nk}^{-1}(z_k - \hat{z}_k) dt$$

$$= \sum_{n=0}^{N-1} \Re T_{nk}^{-1}(z_k - \hat{z}_k) 2 \int_0^1 \frac{\partial f}{\partial x_n} dt$$

$$= 2 \sum_{n=0}^{N-1} \Re T_{nk}^{-1}(z_k - \hat{z}_k) \frac{IG_n}{x_n - \hat{x}_n}$$

Denoting $(z_k - \hat{z}_k) = r_k e^{j\phi k}$ then

$$\Re T_{nk}^{-1}(z_k - \hat{z}_k) = \frac{r_n}{\sqrt{N}} cos\left(\frac{2\pi kn}{N} + \phi_k\right) \quad (19)$$

And finally,

$$R_k = 2r_k \sum cos\left(\frac{2\pi kn}{N} + \phi_k\right) \frac{R_n}{x_n - \hat{x}_n} \quad (20)$$

Which is equivalent to the method of (Vielhaben et al., 2024).

As an example, we present (Figure 6) the frequency attributions of Figure 2, comparing the Cross-Domain IG in the frequency domain with the Virtual Inspection Layer on top of the time-domain IG.

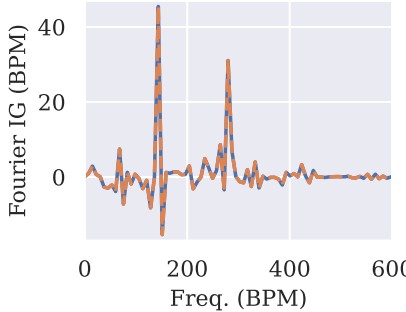

*Figure 6.* Example of Cross-Domain IG in the frequency domain compared to the Virtual Inspection Layer operating on the time-domain IG.

## G. Relation to FreqRISE

We experimentally compared Cross-domain IG (frequency and time-frequncy domains) to FreqRISE (Brüsch et al., 2025a). We run their benchmarking on the AudioMNIST dataset (Becker et al., 2024), evaluating Faithfulness and Complexity as defined by Jang et al. (Brüsch et al., 2025a).

Additionally, we evaluated Cross-domain IG in the Complex Spectrum and Discrete Wavelet (DWT) domains. For the Complex Spectrum the transform $T$ is defined as:

$$T = \mathcal{F}^{-1}\{log(\mathcal{F}\{x(t)\})\} \quad (21)$$

$\mathcal{F}$ is the Fourier transform. For the DWT we used the Haar wavelet decomposition. The results are presented in Table 4.

## H. Feature-level Insertion-Deletion

We perform insertion-deletion evaluation tests on the three examples presented in Section 5.1. Our evaluation indicates that component-level attributions provide more faithful and concentrated evidence for the models' predictions than time-domain attributions: adding top-rated component features rapidly reconstructs the output, while removing them destroys it.

### H.1. Heart rate extraction from physiological signals

We follow the procedure outlined below:

1. **Select** $k\%$ **features**, either in the time or frequency domain. For the frequency and time domain IG, we select the $k$ components with the highest IG score. For the random intervention, we randomly sample $k\%$ unique frequency bins.

2. **Insert/delete** $k$ components to generate modified samples $\boldsymbol{x}_{mod}$.

3. **Infer** heart rate with $\boldsymbol{x}_{mod}$ input.

4. **Compare** $f(\boldsymbol{x}_{mod})$ with the original heart rate inference before any interventions $f(\boldsymbol{x})$.

An example of inference after inserting/deleting input features is presented in Figure 7. We plot the heart rate inference throughout the entire 2-hour session of subject 15 from the PPG-Dalia dataset. The results for the entire PPGDalia dataset are summarised in Table 5.

### H.2. Electroencephalography-based epileptic seizure detection

We used the Physionet Siena Scalp EEG Database v1.0.0 (Detti, 2020; Detti et al., 2020; Goldberger et al., 2000). For each subject's sessions, we retrieved the first sample that is detected as a seizure by the `zhu-transformer`. For each sample, we generated ICA-domain IG saliency maps and performed insertions/deletions with the most important IC. We kept track of the change in the seizure classification probability, $\Delta p = p(\boldsymbol{x}_{mod}) - p(\boldsymbol{x})$, as we:

1. Delete the most important component and perform inference,

2. Maintain the most important component, delete the rest of the components, and perform seizure classification.

|  | Frequency | | Time-Frequency | | DWT | | Complex Spectrum | |
|---|---|---|---|---|---|---|---|---|
|  | **Digit** | **Gender** | **Digit** | **Gender** | **Digit** | **Gender** | **Digit** | **Gender** |
| Faithfulness ↓ | | | | | | | | |
| **FreqRISE** | 0.160 | 0.416 | 0.104 | 0.423 | - | - | - | - |
| **LRP** | 0.205 | 0.431 | 0.214 | 0.420 | - | - | - | - |
| **IG** | 0.252 | 0.428 | 0.197 | 0.389 | - | - | - | - |
| **CDIG (ours)** | 0.19 | 0.446 | 0.099 | 0.429 | 0.254 | 0.603 | 0.097 | 0.505 |
| Complexity ↓ | | | | | | | | |
| **FreqRISE** | 8.17 | 8.01 | 10.82 | 10.78 | - | - | - | - |
| **LRP** | 5.84 | 5.16 | 4.67 | 4.16 | - | - | - | - |
| **IG** | 6.41 | 4.74 | 5.26 | 4.04 | - | - | - | - |
| **CDIG (ours)** | 6.31 | 4.609 | 5.143 | 3.777 | 1.300 | 1.309 | 4.219 | 4.063 |

*Table 4.* Comparison of Cross-domain IG with FreqRISE. LRP and IG refer to the use of a Virtual Inspection Layer (Vielhaben et al., 2024) on top of the time-domain LRP and IG, respectively. The faithfulness and complexity scores for FreqRISE, LRP and IG are taken from (Brüsch et al., 2025a). Since FreqRISE, LRP and IG are instantiated only in the frequency and time-frequency domains, they cannot provide saliency maps in the Complex Cepstrum domain. We have also added evaluations of Cross-Domain IG in the Discreate Wavelet Transform.

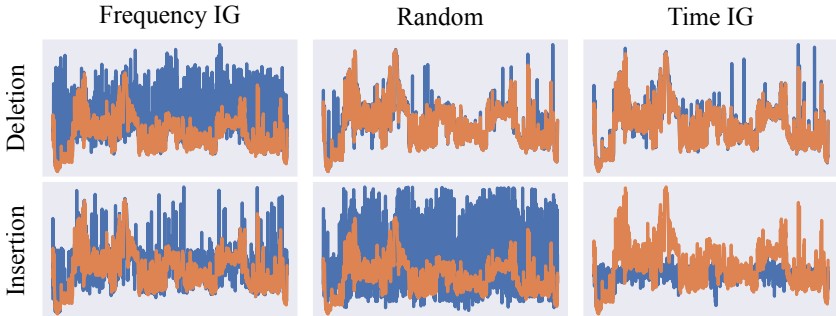

*Figure 7.* **Example of heart rate inference after deleting features.** We plot the entire session of subject 15 from PPGDalia. For each insertion/deletion, we retain/delete 3.125% of the input features. For the Fourier and time IG these are the frequency bins and time-points with the highest assigned IG score. In the random case, we randomly drop 3.125% of the frequency bins. We plot the **original HR inference** over the duration of the session and the model's output after **modifying the input** accordingly.

We compared these results with those obtained from randomly choosing an IC component and performing the same insertion/deletion evaluation.

## I. Example time-domain attributions

Figures 8, 9 and 10 present the time-domain attributions from the examples of Section 5.1. In all three cases, interpreting the time-domain saliency maps is difficult and of limited utility.

**Heart rate inference.** The time-domain IG highlights individual time-points of the PPG input. However, it is difficult to assess:

1. *Does an individual time-point contribute to the heart or interference components?* In the time domain, both the effect of the heart and the interference are mixed, and each time point contains information from both of

these components. In contrast, in images, when there is component (object) overlap, one component blocks the other, and a single pixel carries single-component information.

2. *Which time-points should be the most important/influential?* From domain knowledge we know that oscillations around the ground truth heart rate should be the ones affecting the model's output. However, we do not have any such insights in the time domain, and the component overlap further complicates oscillation identification in time.

Consequently, these saliency maps do not allow us to answer the interpretability task of Section 5.1.1.

**Seizure detection.** Similarly to the heart rate example, it is not easy to visually identify the seizure-related oscillations in the time-domain saliency map.

| Top k%-features | 3.125 % | 25% | 50% |
|---|---|---|---|
| Deletion ↑ | | | |
| Frequency IG | 66.39 | 133.56 | 127.13 |
| Time IG | 10.13 | 50.86 | 104.84 |
| Random | 8.53 | 37.03 | 68.34 |
| Insertion ↓ | | | |
| Frequency IG | 37.98 | 20.08 | 9.86 |
| Time IG | 94.58 | 57.27 | 58.61 |
| Random | 123.71 | 100.39 | 66.67 |

*Table 5.* Insertion-deletion evaluation dropping the k% most important features. Deletion/Insertion distance (expressed in Beats per Minute- BPM) from the original HR inference averaged across 15 subjects of PPGDalia.

| | ICA IG | Random |
|---|---|---|
| Deletion $\Delta p$ ↑ | 0.1776 | 0.0083 |
| Insertion $\Delta p$ ↓ | 0.0696 | 0.4396 |

*Table 6.* Insertion-deletion evaluation on the seizure detection model.

**Time series forecasting.** The time-domain IG highlights mostly the last input time-points.

## J. Additional examples

We present additional Cross-domain IG examples in Figures 11, 12 and 13.

## K. Baseline sensitivity

CDIG inherits the baseline dependence of IG. In the main experiments, we use a zero-signal baseline for consistency across domains and because, in our settings, it corresponds to an absence-of-signal reference in the original input space. To assess sensitivity to this choice, we repeat the Table 2 benchmark with the dataset mean as an alternative neutral baseline. The results are shown in Table 7.

The mean-baseline experiment supports two conclusions. First, CDIG is not tied to a single hand-picked baseline: under the mean baseline, Cepstrum remains above TIMEX++ on all five datasets under both CPD protocols, while Fourier does so on nine out of ten dataset/protocol pairs. Second, the baseline materially affects attribution values and can change the relative ordering between domains. We therefore view the baseline as part of the interpretability setup, rather than as a CDIG-specific nuisance parameter.

## L. Robustness to input perturbations

We evaluate the stability of CDIG under additive input perturbations in both a controlled synthetic setting and a real-world PPG setting.

**Synthetic Fourier setup.** We first use the tractable setup from Section 4.2. In the ideal single-component case, frequency-domain CDIG concentrates attribution at the active frequency and scales with the signal amplitude and filter gain. Under additive noise, we therefore expect attribution mass to spread progressively to off-target frequencies as SNR decreases, rather than fail abruptly. Because the model includes a ReLU nonlinearity, this reasoning is approximate under noise. We verify it empirically by measuring the fraction of attribution mass retained in the target frequency support (Table 8).

The attribution mass remains concentrated around the target frequency at high and moderate SNRs, and degrades gradually as the perturbation becomes dominant.

**PPG robustness under noisy inputs.** We also evaluate robustness on the PPG heart-rate extraction task from Section 5.1.1. We add random time-domain noise and compare clean and noisy saliency maps. To isolate attribution stability from trivial prediction drift, we restrict the analysis to samples whose predicted heart rate remains within 5 BPM of the clean prediction. We report the Pearson correlation between clean and noisy saliency maps for both time-domain IG and Fourier-domain CDIG (Table 9).

In this setting, Fourier-domain CDIG remains substantially more stable than time-domain IG under realistic perturbations. The top-1 attributed frequency bin also remains unchanged between clean and noisy inputs, consistent with the PPG interpretability task, where the explanatory structure is frequency-based rather than pointwise in time.

## M. EEG and ICA

The raw EEG input is presented in Figure 14.

The implementation of the `zhu-transformer` we used can be found here https://github.com/esl-epfl/zhu_2023.

The application of ICA in EEG signals is based on the general assumption that the EEG data matrix $X \in \mathbb{R}^{N \times M}$ is a linear mixture of different sources (activities) $S \in \mathbb{R}^{N \times M}$ with a mixing matrix $A \in \mathbb{R}^{N \times N}$ such that $X = AS$, where $N$ is both the number of sources and EEG channels, and $M$ is the number of samples in the dataset. Sources are assumed to be statistically independent and stationary. These assumptions can be leveraged to compute an inverse unmixing matrix $W = A^{-1} (\in \mathbb{R}^{N \times N})$, such that $S = WX$.

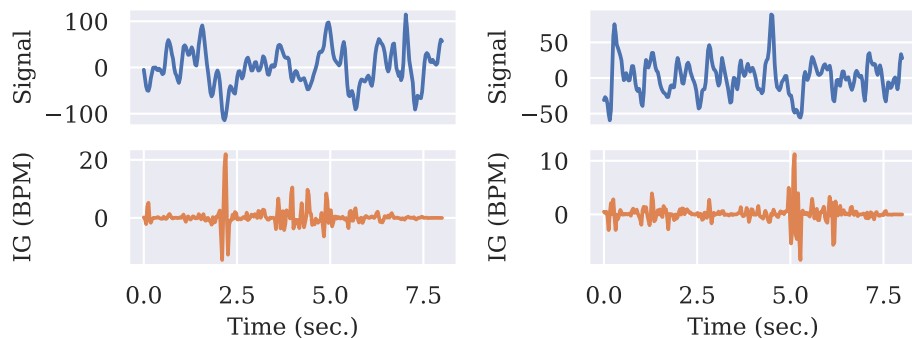

*Figure 8.* **Time-domain IG for HR inference**. We present the same two inputs as in Figure 2. For each time point in the input we assign a significance value. **Top:** Raw time-domain input which is processed by the model. **Bottom:** IG saliency map expressed in the original time domain.

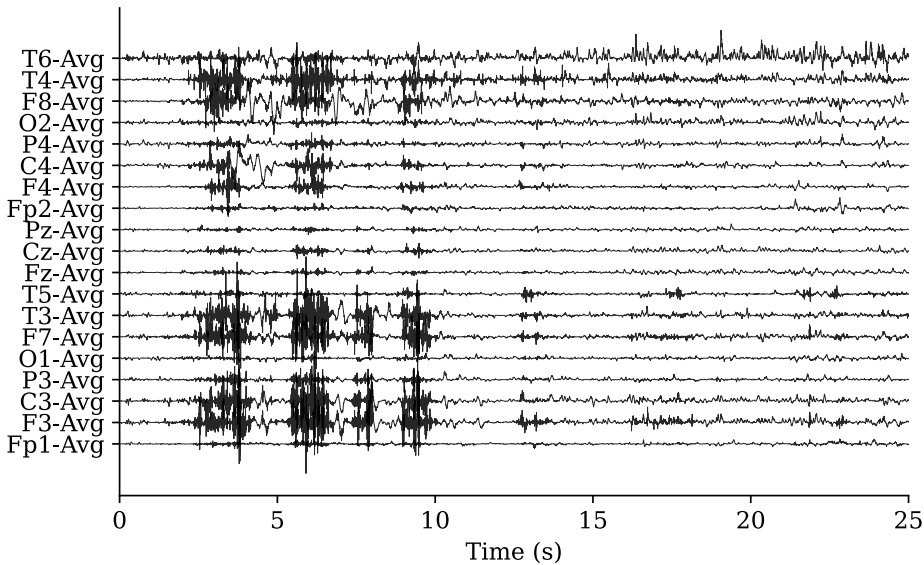

*Figure 9.* **Time-domain IG for seizure classification**. For each time point on each channel we assign a significance value.

|  | PAM | | Boiler | | Epilepsy | | Wafer | | Freezer | |
|---|---|---|---|---|---|---|---|---|---|---|
|  | Avg. | Zero | Avg. | Zero | Avg. | Zero | Avg. | Zero | Avg. | Zero |
| Fourier | 0.047 | 0.436 | 0.782 | 0.995 | 0.140 | 0.143 | 0.340 | 0.340 | 0.273 | 0.273 |
| Cepstrum | 0.386 | 0.371 | 1.31 7 | 0.390 | 0.226 | 0.709 | 0.826 | 0.812 | 0.330 | 0.431 |

*Table 7.* CPD results for Fourier- and Cepstrum-domain CDIG using the dataset mean as the IG baseline. The "Avg." and "Zero" columns refer to the CPD substitution protocol, not the IG baseline. Higher is better.

| SNR (dB) | 20 | 10 | 5 | 0 | -5 | -10 | -20 |
|---|---|---|---|---|---|---|---|
| MR | 0.96 | 0.95 | 0.89 | 0.87 | 0.73 | 0.46 | 0.10 |

*Table 8.* Robustness of frequency-domain CDIG in the synthetic Fourier setup under additive noise. MR denotes the fraction of attribution mass retained in the target frequency support.

Finding $W$ is an ill-posed problem without an analytical solution, which can be estimated by means of different ICA algorithms (Hyvärinen et al., 2001; Klug and Gramann, 2021). ICA is used in EEG to decompose the signal into independent components that separate the signal of interest from various sources of artifacts (Winkler et al., 2011). In this work, for ICA we selected the FastICA algorithm implemented in `sklearn` (`max_iter` = $3 \cdot 10^4$, `tol` = $1 \cdot 10^{-8}$).

The independent channels estimated using ICA are pre-

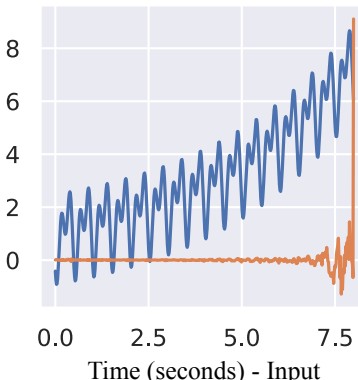

*Figure 10.* **Time-domain IG for time-series forecasting**. We plot the raw time-domain **input** along with the **IG importance** for each time-point in the input.

| SNR (db) | 20 | 10 | 5 |
|---|---|---|---|
| Time-domain IG | 0.72 | 0.38 | 0.21 |
| Fourier-domain CDIG | 0.92 | 0.80 | 0.66 |

*Table 9.* Clean-vs-noisy attribution stability on the PPG task. Pearson correlations are computed between saliency maps from clean and noisy inputs, restricted to samples whose noisy prediction remains within 5 BPM of the clean prediction.

sented in Figure 15.

## N. Generated time series for TimesFM forecasting

We generate a synthetic time series signal, $x(t)$, composed of an exponential trend, $x_{trend}(t)$, and a seasonal component, $x_{seasonal}(t)$:

$$x_{trend}(t) = e^{\frac{t}{\alpha}}$$
$$x_{seasonal}(t) = sin(2\pi \cdot \xi \cdot t + \phi) + sin(2\pi \cdot 2\xi \cdot t + \phi)$$
$$x(t) = x_{trend}(t) + x_{seasonal}(t)$$

For the example in Section 5.1.3 $\alpha = 4$, $\xi = 2Hz$. For the samples presented in Appendix J they were randomly sampled from $\alpha \sim U(4.0, 7.0)$ and $\xi \sim U(3.0, 8.0)[Hz]$. A window of 512 time points, starting at $t = 0$, are given as input to TimesFM which generates forecasts up to 128 time points in the future from $t = 512$. The input time series and STL decomposition are presented in more detail in Figure 16.

## O. Limitations

In this work, we have addressed the limitations of IG regarding time-domain saliency maps. The rest of the original IG limitations are also transferred to our method. For example, the current implementation focuses on a linear integration path, reflecting the original IG. However, other non-linear

paths, e.g., Guided IG (Kapishnikov et al., 2021), should be explored. In our Remarks in Section 4 we briefly note how already available solutions to these limitations could be transferred directly to Cross-domain IG. For clarity, we summarise them here:

1. **Integration path.** In this work, we used a linear path in line with the original IG (Sundararajan et al., 2017). However, eq. 5 allows for the use of non-linear curves such as in (Yang et al., 2023; Kapishnikov et al., 2021).

2. **Choosing the baseline.** In (Sundararajan et al., 2017) the authors argue that a baseline point exists for most deep networks. In cross-domain IG, if such a point exists, then it can be trivially defined in the target domain through the transform $T$.

3. **Computational overhead.** Similarly to the original IG, our method requires multiple differentiations to approximate the integral (Definition 4.1). We require an additional step due to the transform $T$: computing the inverse and the backpropagation over it.

Our method requires an invertible, differentiable transform and a carefully selected baseline point. Consequently, we excluded non-invertible transforms, and further investigation is needed for approximately invertible cases. Baseline selection also plays a role in the final saliency map. We focused on the zero-signal as the baseline point; future work should include an extensive investigation into the effects of the baseline selection. Finally, multiple transforms can be combined to provide a multi-faceted saliency map, such as ICA combined with frequency domains, and automatic transform selection could help streamline the process. We leave *ensemble* domains and automatic domain selection as future work.

## P. Experiments compute resources

All experiments were run on an NVIDIA Tesla V100 with 32 GB of memory.

## Q. Use of LLMs

We used a large language model (LLM) for light copy-editing (grammar and wording) and minor coding assistance (e.g., debugging errors).

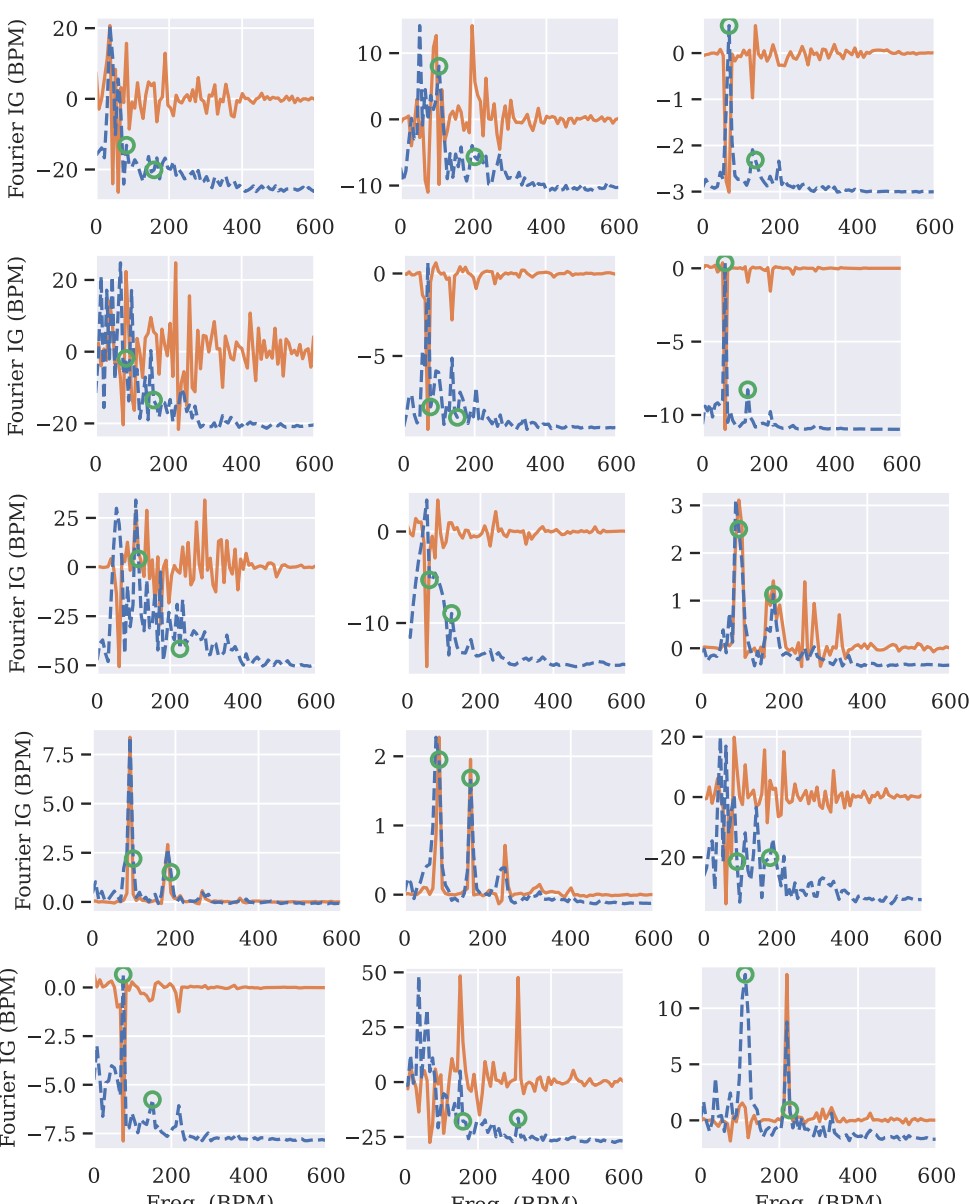

*Figure 11.* **Frequency-domain IG for heart rate inference model.**

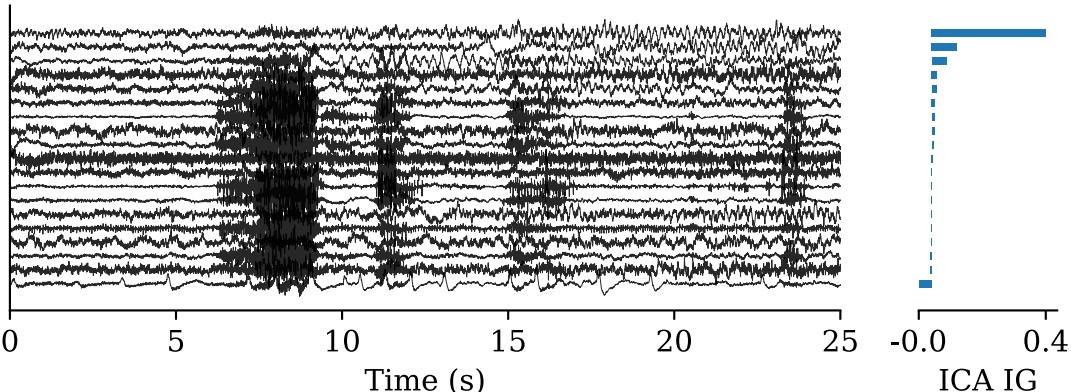

*Figure 12.* **ICA-domain IG for seizure detection model.** Similarly to the example presented in Section 5.1.2, the first channel contains the majority of the seizure components. IC channels that contain mostly interference are assigned a very small IG score.

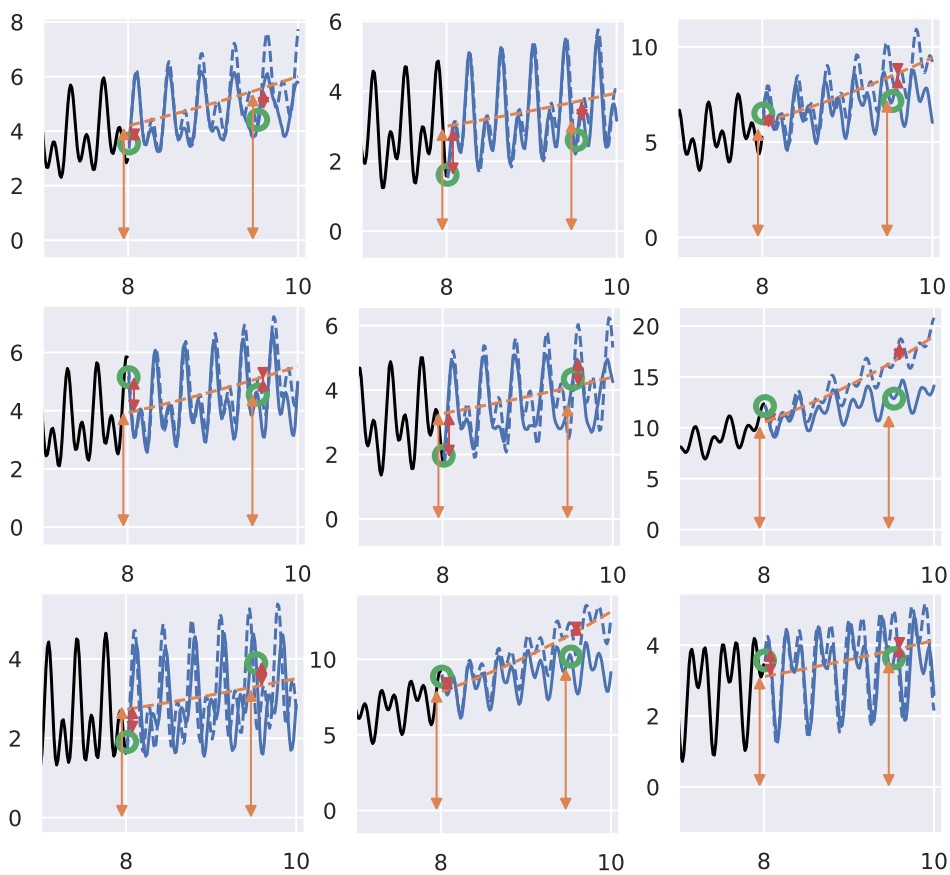

*Figure 13.* **Seasonal-Trend IG for TimesFM forecasts**. We generate synthetic samples by sampling them as described in Appendix N.

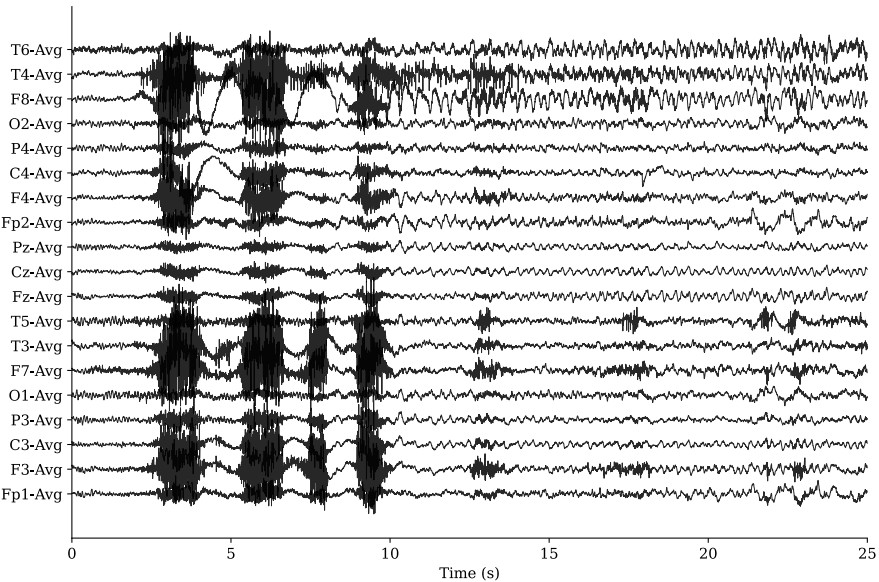

*Figure 14.* EEG signal in the original channel space.

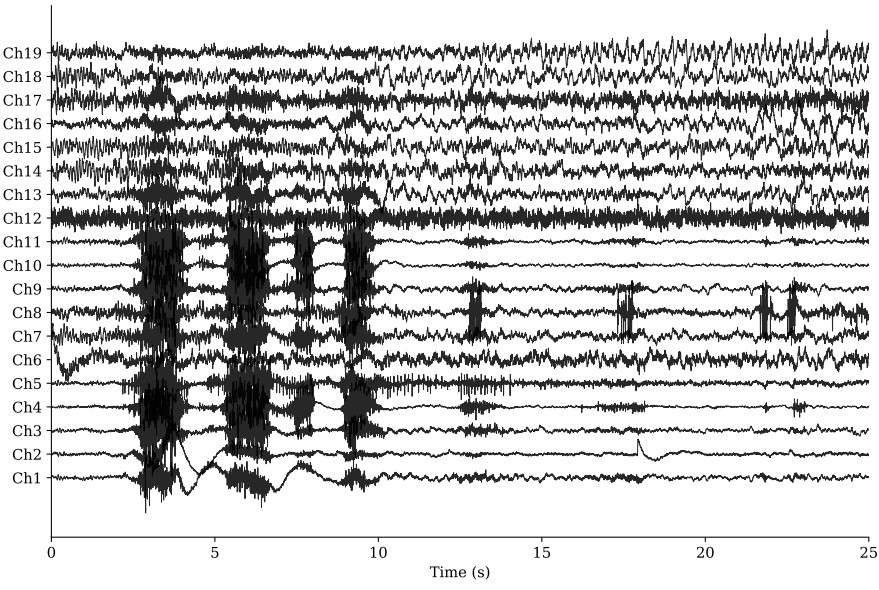

*Figure 15.* EEG signal in the Independent Component space.

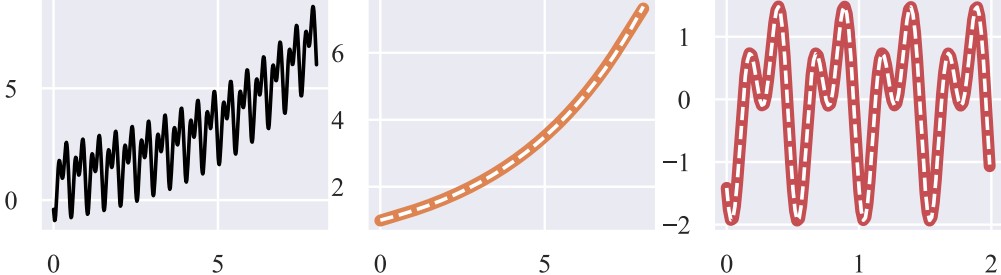

*Figure 16.* **Input time series for forecasting and successful STL decomposition. Left:** time series with a **trend** and a **seasonal** component. **Center:** The decomposed **trend** component and ground truth trend (white dashed line). **Right:** The decomposed **seasonal** component and ground truth seasonality (white dashed line).

