# OpenReview forum: "Time series saliency maps: Explaining models across multiple domains"
_ICML.cc/2026/Conference — ICML 2026 spotlight_

### Official Review · Reviewer_RFyj · 2026-03-12

**Soundness:** 2
**Presentation:** 3
**Significance:** 3
**Originality:** 3
**Overall Recommendation:** 5
**Confidence:** 4

**Summary:**

This paper presents cross domain integrated gradients, a saliency map method that allows for feature attributions in any domain that can be formulated as a differentiable transformation of the time domain. They deploy cross-domain IG on multiple time series applications and models. The authors release their implementation as an open source tensorflow pytorch library.

**Compliance With Llm Reviewing Policy:**

Affirmed.

**Final Justification:**

Strong paper with high significance, originality, good presentation and soundness. The authors addressed my minor concerns surrounding method sensitivity, reliability and robustness and scalability, so I raised my score from a 4 to a 5.

**Key Questions For Authors:**

1.	How sensitive is this method to noise and input perturbations? Are the resulting attributions stable under realistic levels of variability?
2.	How reliable and robust are the generated saliency maps in practice? Was this measured quantitatively in any way?
3.	What is the performance overhead for cross domain IG? How scalable is the method? How does it scale with time‑series length, sample frequency, dataset size, or domain choice?

**Limitations:**

yes

**Strengths And Weaknesses:**

Significance & Originality: This paper targets the important area of time series explainability, offering a meaningful advancement in the field. The core contribution of generalizing Integrated Gradients (IG) to any domain that is differentiably transformed from the time domain fills a clear gap in current time series saliency methods, which often struggle to capture informative latent structures. The authors demonstrate the breadth of their method by instantiating it across six different domains and validating its utility through a series of case studies aligned with varied interpretability goals. They also evaluate the method using multiple model architectures, including deep neural networks and foundation models, and prediction tasks which helps support the general applicability of the approach. The release of an open source tensorflow/pytorch package is commendable.

Presentation: The paper is well written, well motivated, and easy to follow. The mathematical definitions are precise and the graphics and figures are clear. The discussion in Section 3.2 is particularly insightful.

Soundness: The proposed method is sensible and appears technically sound. The theoretical formulation is valid and the experiments provide useful evidence of effectiveness. That said, there are several weaknesses that, if addressed, would significantly strengthen the overall impact of the work.

Weakness 1: Like other gradient‑based saliency approaches, the method may be susceptible to noise, unstable under small input perturbations, or prone to variability across similar samples. See question 1. Saliency methods are also known to highlight irrelevant regions or exhibit artifacts. The paper does not quantitatively evaluate robustness, making it difficult to assess reliability and trustworthiness in practice. See question 2.

Weakness 2: A major missing component is the absence of an analysis of computational or performance overhead. As is common with saliency methods, significant costs may be incurred depending on the transform, time series length, sampling rate or number of traces. An evaluation of scalability would strengthen the practical relevance of the work. See question 3.

Weakness 3: The approach assumes that practitioners already have some intuition about which transform domain (e.g., ICA, DWT, Fourier) is most appropriate for a given application. This may limit usability. Additionally, the method only applies to invertible, differentiable transforms, which constrains applicability in settings where desirable transforms do not meet these requirements. The authors acknowledge these limitations themselves; I note that I do not penalize the paper for this category of weaknesses, but do mention them here since they are noteworthy drawbacks.

---

> ### Author Rebuttal · Authors · 2026-03-30
>
> We thank the reviewer for highlighting robustness and scalability as important dimensions of trust for saliency methods. To address these concerns directly, we ran additional perturbation and runtime analyses, and we summarise them below.
>
> Q1. We ran additional additive-noise experiments.
>
> **Synthetic Example.** In our tractable setup from Section 4.2, frequency-domain CDIG has a clear expected behaviour under perturbation: in the ideal single-component case, attribution is concentrated at the active frequency and scales with signal amplitude and filter gain (eq. 6). Under additive noise, one would therefore expect attribution mass to spread progressiveley to off-target frequencies as SNR decreases rather than fail abruptly. Because the model includes a ReLU, this reasoning is only approximate under noise, so we verified it empirically by measuring the fraction of attribution mass retained in the target frequency support (MR):
>
> |**SNR(dB)**|20|10|5|0|-5|-10|-20|
> |-|-|-|-|-|-|-|-|
> |MR|0.96|0.95|0.89|0.87|0.73|0.46|0.10|
>
> **Real-world example.** We added random time-domain noise and compared clean vs noisy attributions on the PPG task (Section 5.1.1). To isolate attribution stability from trivial prediction drift, we restricted analysis to samples whose predicted hear rate remained within 5 BPM of the clean prediction. We measured Pearson correlation between clean and noisy saliency maps for both time- and fourier-domain IG. :
>
> | **SNR(dB)**|20|10|5|
> |-|-|-|-|
> | **Pearson(time)**|0.72|0.38|0.21|
> | **Pearson(freq.)**|0.92|0.80|0.66|
>
> Thus, in the PPG setting, Fourier-domain CDIG remains markedly more stable than time-domain IG under realistic perturbations. The top-1 attributed frequency bin also remained unchanged between the clean and noisy inputs, consistend with the PPG interpretability task, where the explanatory structure is frequency-based rather than pointwise in time.
>
> Q2. Beyond the perturbation analysis above, the submission already includes quantitative and mechanistic evaluations of reliability.
>
> First, we evaluate whether the features identified by CDIG materially affect the model output in the expected direction. On PPG and EEG, deleting the top-k CDIG features affects the model output considerably more than in time-domain or with random feature deletion. We observed corresponding behaviour of higher faithfulness in insertion tests. In the benchmark setting, CDIG is reported as consistently stronger than TIMEX++ and competitive with TIMING under CPD.
>
> Second, the paper includes mechanistic validation. In Section 3.2/Figure 1, we analyse a tractable CNN whose mechanism is known. There, time-domain IG is difficult to interpret mechanistically, whereas frequency-domain CDIG highlights exactly the frequency components that actually drive the prediction. Section 4.2 strengthens this analytically, deriving a closed-form link between the complex-domain attribution and the filter's frequency response. The new perturbation results (Q1) are complementary: in a setting where the model's decision mechanism is known, CDIG aligns with that mechanism, and under realistic perturbations on PPG is saliency remains more stable than time-domain IG.
>
> Finally, as in standard IG, CDIG preserves the same underlying attribution principles (e.g. completeness), but in a semantically meaningful transformed domain.
>
> Q3. We ran additional runtime experiments comparing standard IG and Fourier-domain CDIG on CNNs with fixed width (64 channels). These isolate the extra cost from the transform/inverse-transform step. CDIG adds a measurable but modest overhead, and this overhead becomes a smaller fraction of total attribution time as model cost increases.
>
> **Const. Input size (256)**
> |**Depth**|3|6|9|12|
> |-|-|-|-|-|
> |\% Ovhd|27.68|12.09|8.93|10.10|
>
> **2 Layers**
> |Input|256|512|1024|2048|4096|
> |-|-|-|-|-|-|
> |\% Ovhd|23.65|24.99|21.29|15.14|11.54|
>
> **9 Layers**
> |Input|256|512|1024|2048|4096|
> |-|-|-|-|-|-|
> |\% Ovhd|10.98|8.63|6.21|3.42|1.77|
>
> These results suggest that additional transform-related cost is most noticeable for small models/short inputs, while for larger models the forward backward cost through the network dominates. This is consistent with the manuscript's discussion that CDIG is computationally similar to IG, with extra overhead coming from the inverse-transform step. For domains such as ICA, there can be an upfront cost to construct the representation itself, but this cost is not specific to CDIG; it would be required by any explainer operating in that domain. The CDIG-specific overhead is the additional differentiation through the fixed transform/inverse-transform layer once the representation has been defined.
>
> **Transform choice.** More broadly, we agree that transform choice matters. Our view is that the explanation domain should be chosen from the interpretability question and signal semantices (e.g., frequency for PPG), rather than selected post hoc to fit a desired narrative (see also Q1 4xpx).

---

> > ### Author Rebuttal · Reviewer_RFyj · 2026-03-31
> >
> > Thanks for your detailed response. My questions and concerns have been fully addressed and I will raise my score.

---

### Official Review · Reviewer_Kqv4 · 2026-03-12

**Soundness:** 3
**Presentation:** 3
**Significance:** 3
**Originality:** 3
**Overall Recommendation:** 5
**Confidence:** 2

**Summary:**

The authors extend IG to the a complex input domain. This allows them to compute input attribution over a wide range of arbitrary transformations of the original time-series datapoints, as long as this transformation is invertible and differentiable. This tackles a common difficulty in input attribution of time series models: Neighbouring timepoints may be semantically disjoint. Individual time-points do not offer much explainability as in time-series predictive factors usually lie in latent features, like frequency or disentangled data sources. By attributing using IG to a transformation of the original time series datapoints these latent features can be made apparent, such as DFT for frequency and ICA for splitting emission sources. The authors provide a sound mathematical basis for their methods, showing path independence and maintaining completeness as in time-domain IG. The authors evaluate their methods in three ways: (1) A theoretical argument using a minimal single-layer CNN, (2). a qualitative evaluation on 3 different transformations, and (3) a quantitative evaluation via feature level insertion/deletion of identified key features by their method on 2 domains. The authors highlight that CDIG suffers from most of the same limitations as time-domain IG.

**Compliance With Llm Reviewing Policy:**

Affirmed.

**Final Justification:**

Thank you for your response. All my concerns have been addressed. I have updated my score to accept.

**Key Questions For Authors:**

* What would be a good zero-information baseline in frequency space?
* How were baselines chosen in the experiments?
* Would you consider adding your code to captum?

**Limitations:**

Yes

**Strengths And Weaknesses:**

**Soundness**

The mathematical derivation of CDIG is thorough and correct. The preliminaries section is appreciated. Minor mistake in line 233, 234 (rightside) b is first indexed with $j$, later with $I$.

The authors quantitatively compare their methods with related work where possible. Table 1 shows the versatility of CDIG. In my opinion that is it’s core strength. The quantitative results showed that CDIG achieves comparable faithfulness and computational complexity to VIL when operating in the frequency domain. CDIG is a generalized approach that successfully generates saliency maps in other transform domains, such as the Discrete Wavelet Transform and the Complex Cepstrum, which VIL does not support.

Minor: Appendix G mentions 3 insertion-deletion experiments, but only two are described (Sec 5.2 of main text also only mentions running experiments on two real-world problems).

**Presentation**

I wonder if the title is truly fitting. Across multiple domains suggests the method actually provides explainability ACROSS multiple domains in a multi-view setting. Where as in reality the practitioner has to select a specific domain themselves and the method provides a saliency map only in that domain. A name like complex integrated gradients would be more fitting.

The abstract is too long as per ICML guidelines.

Citation style should be corrected. Please use \citet or \citeauthor for inline citations and \cite for end of line citation.

Minor notation suggestions: I completely understand that a researcher, in describing mathematical theory rigorously, sometimes finds themself running out of alphabet, but some notation conventions are best left honoured. My suggestions would be to reserve $i$ (or $j$ is you must) for indicating the imaginary part of complex numbers and use other letters for indexing. This avoids confusion for the reader, as well as prevent mistakes from slipping in as mentioned above (line 233-234).

**Significance**

This is where I have my main concerns. CDIG suffers from many of the same limitations as time-domain IG, as is mentioned by the authors. Specially selecting the right baseline is a well known issue for IG-methods. Remark 4.5 addresses this for CDIG, but does not provide any corroboration or additional evidence that takes away this concern. For example, what would be a good zero-information baseline in frequency space?

CDIG leaves selecting the transformation domain to the user. This requires prior domain knowledge and has a potential to encourage domain-shopping or confirmation bias to fit a specific explanation. As the authors state, CDIG may produce clean-looking saliency maps that may be semantically misleading. The paper does not provide a strategy to address this issue, nor a clear enough warning to potential practitioners.

The CDIG methodology fundamentally constraints the saliency map to a single domain (a single view). While elegant, this limits the usefulness of the explainability method. By forcing the explanation into one specific, globally transformed domain (like DFT or ICA), CDIG may successfully isolate one type of latent structure but completely obfuscate others. It lacks any interactive mechanism (like MIX's cross-view refinement) to dynamically evaluate and aggregate features across different time-frequency resolutions.

The authors provide an open-source implementation of their method. This is useful to the community and allows practitioners to easily test the usefulness of CDIG for themselves in a particular use case.

**Originality**

Generally I think the idea of adding a transformation before input attribution to adjust the specific view domain for time-series saliency maps is good.The authors expand this paradigm by mathematically showing how IG can be made to support complex transformations of the input domain. This is especially useful for transformations dealing with frequencies. I believe the authors idea is original and useful. The question I have is whether the work is distinct enough from related work proposing a similar core idea of transforming the view domain of time-series saliency maps. I trust other reviewers more familiar with the subfield to judge the novelty of this contribution.

---

> ### Author Rebuttal · Authors · 2026-03-30
>
> Q1. We agree that this is an important point. In our formulation, the baseline is defined in the original signal space and then mapped through the transform: if $\hat{x}$ is the signal-space baseline, the corresponding transform-domain baseline is $\hat{z} = T(\hat{x})$. This is the principle formalised in Remark 4.5: if $\hat{x}$ is a valid reference point for the original model and $T$ is invertible, then $\hat{z}$ is immediately defined in the transformed-input formulation. Thus, in the Fourier case, a zero-signal baseline induces the zero spectrum. More generally, IG does not require $f(\hat{z}) = 0$. The baseline defines the reference against which contributions are decomposed. A neutral baseline answers an "absence-of-information" question, while other baselines can answer contrastive questions. In CDIG, the key principle is therefore to choose the baseline according to the underlying signal-level interpretability question, and then induce the corresponding reference in the chosen transform domain, rather than selecting a baseline ad hoc in transformed coordinates.
>
> Q2. To clarify the current submission, all experiments used the zero-signal baseline for both IG and CDIG. We made this choice for three reasons:
> 1. it follows the standard neutral-reference logic used in IG.
> 2. it keeps the empirical study controlled by avoiding an additional source of variability across tasks and transform domains, and
> 3. our results suggest that this choice is sensible across the settings we study: in the mechanistic Fourier example it induces the natural zero spectrum and yields the analytic link to filter frequency response, while in the real-world evaluations CDIG with this baseline produces quantitatively faithful attributions.
> Existing distribution-of-baselines variants of IG are directly compatible with our formulation but were ousider the scope of the present work.
>
> To further assess sensitivity to this choice, we also repeated the Table 2 benchmark experiments with the dataset mean, $\mathbb{E}[x]$, as an alternative baseline (see response to XA1h Q1). The main conclusions remain qualitatively similar in that setting, although the baseline does affect attribution values and can change the relative ordering between domains. We will clarify this more explicitly in the final version.
>
> Q3. Yes, we agree that Captum would be a natural fit for the PyTorch implementation and would improve accessibility for practitioners already using that ecosystem. We will add this integration to our codebase.
>
> **On the across multiple domains/single-view concern.** You are right that the current method produces a saliency map in one explanation domain at a time. It is not a multi-view aggregation method in the sense of MIX. Our intended claim is generality across possible explanation domains, not simultaneous cross-domain aggregation. In settings where several complementary decompositions are meaningful, CDIG can be instantiated separately under pre-specified transform choices to provide complementary analyses of the same prediction. We did not present this as a central claim because the paper focused on establishing the single-domain formalism and its case studies. We agree that principled multi-view aggregation across such domains is a natural future direction, and we will clarify this distinction more explicitly in the final version.
>
> **On domain-shopping/confirmation bias.** We fully share this concern. The paper's position is not that any domain yielding a visually clean map should be trusted. On the contrary, a poor domain choice can produce semantically misleading attributions even when the map appears plausible. Our intended usage protocol is: first specify the interpretability question. Then, choose a transform whose coordinates correspond to the scientifically meaningful factors for that question; fix that transform before inspecting saliency maps. When multiple plausible transforms exist, treat them as complementary sensitivity analyses rather than selecting the most visually appealing one. We will make this warning and protocol substantially more explicit in the final version.
>
> **On comparisons to related methods.** Our goal is not to claim that CDIG uniformly outperforms specialised methods in every setting. Rather, the contribution is a transform-agnostic framework together with a principled extension of IG to complex-valued transform domains with IG-style guarantees. Where direct frequency-domain comparisons are available, the paper reports competitive faithfulness/complexity, while also extending naturally to additional transform families beyond domain-specific alternatives.
>
> **Presentation comments.** We also thank the reviewer for the presentation comments. The Appendix G mismatch is a drafting error on our side: only two real-world insertion/deletion evaluations were run. We also appreciate the comments on the abstract length, notation, and citation style. We will address these in the final version.

---

> > ### Author Rebuttal · Reviewer_Kqv4 · 2026-04-01
> >
> > Thank you for the detailed response. My primary concerns regarding baseline selection and the potential for confirmation bias have been adequately addressed. Your proposed protocol, requiring practitioners to pre-specify the transform based on the interpretability question prior to inspecting the map, is an okay mitigation strategy. Please ensure this warning and protocol are prominently featured in the final text.
> >
> > Regarding baseline selection: Conversely, if it were easier design the baseline in the transformed domain, is it possible to just define the baseline $\hat{z}$ directly? Because the transform is strictly invertible, you just apply the inverse transform $\hat{x}=T^{-1}(\hat{z})$ to generate the required time-domain signal.
> >
> > If so, this is a potential great plus to the method. If not, this raises a potential new limitation.

---

> > > ### Author Response · Authors · 2026-04-02
> > >
> > > Thank you for this helpful clarification question. Yes, if the baseline is more naturally specified in the transformed domain, it can be defined there directly, and then mapped back to the model input via the inverse transform, i.e. $\hat{x} = T^{-1}(\hat{z})$.
> > >
> > > So we agree with your reading: this is a practical plus of CDIG, not a new limitation. Because CDIG assumes an invertible transform, specifying the reference as $\hat{x}$ in the original domain or as $\hat{z}$ in the transformed domain are simply two equivalent ways of defining the same baseline.
> > >
> > > The key point, in our view, is that the baseline should be chosen based on the semantics of the interpretability question, not based on which coordinate system is used to write it down. For example, the reference may correspond to absence of signal, an expected signal, a contrastive reference, or an isolated source/component. CDIG makes it possible to express that same reference in whichever domain is more natural, and then map it consistently to the model input through invertibility. This is fully consistent with the protocol we described in the rebuttal: the interpretability question should determine the transform and the baseline in advance, and these choices should be fixed before inspecting saliency maps rather than selected post hoc.
> > >
> > > We think this can be especially useful in settings where the semantically meaningful reference is awkward to specify in raw time/channel space but natural in transformed coordinates. The EEG/ICA case is one example: the observed EEG channels are mixtures of latent sources, whereas ICA provides source-level coordinates (e.g. eye blinks). If the explanatory question concerns a specific source or artifact, it may be much easier to specify that reference in ICA space and then map it back to the model's input via $T^{-1}$.
> > >
> > > To avoid ambiguity, we will make two points explicit in the final version: (1) in the current submission, all experiments use the same zero-signal baseline for both IG and CDIG, and (2) more generally, the formulation also permits direct specification of the baseline in the transformed domain whenever that is the clearest way to express the intended semantic reference.

---

### Official Review · Reviewer_XA1h · 2026-03-12

**Soundness:** 3
**Presentation:** 3
**Significance:** 3
**Originality:** 2
**Overall Recommendation:** 4
**Confidence:** 4

**Summary:**

The paper suggests Cross-domain Integrated Gradients (CD-IG), an extension of Integrated Gradients for the case when the explanation is based not in the time domain, but in a different representation of the signal, if it is reversible and differentiable. The authors consider several examples of such representations: the frequency domain, ICA components, and trend/seasonal decomposition. The basic idea is that such spaces often provide more interpretable explanations for time series than attributions based on individual time points. The method is demonstrated on the tasks of PPG, EEG and forecasting analysis.

**Compliance With Llm Reviewing Policy:**

Affirmed.

**Key Questions For Authors:**

1. How resilient are the results to the choice of baseline?
2. To what extent is the gain of CD-IG due to the method itself, and to what extent is the choice of domain-specific transform?
3. How does CD-IG compare with existing cross-domain / frequency-domain explainability methods in a direct empirical comparison?

**Limitations:**

The technical limitations of the method are indicated, but it would be worth discussing more clearly that such explanations do not guarantee a causal interpretation and may create excessive confidence in the model.

**Strengths And Weaknesses:**

Soundness

The work raises an important problem of time series interpretation, where the explanation in the time domain often does not agree well with the substantive meaning of the data. The idea of transferring the IG to another feature space seems logical and technically sound. The advantage is that the authors consider several different application scenarios, rather than one particular example.
The main limitation of the method is the need for a reversible and differentiable transformation, which significantly narrows its applicability. In addition, the quality of the explanations strongly depends on the choice of baseline and the representation of the signal itself, and this issue is not fully worked out in the article. The experimental part looks uneven: there is quantitative verification for some tasks, but some of the conclusions remain mostly qualitative. Therefore, the work shows the usefulness of the approach, but does not prove its universality.
In general, the work looks technically correct, and the basic idea is well-founded.

Presentation

The work is generally well written.

Significance

This is an important problem, especially for medical and other applied time series. There is significance, but it is more specialized than broad.

Originality

There is a novelty both in the formalization of the transfer of IG to another area, and in the application of this idea to several types of representations. But this is more a neat extension of an existing approach than a fundamentally new paradigm.

---

> ### Author Rebuttal · Authors · 2026-03-30
>
> Q1. We agree that baseline choice is important, since CDIG inherits the baseline dependence of IG. In our paper submission we used a zero-signal baseline throughout for consistency across domains and because, in our settings, it corresponds to an absence-of-signal reference in the original input space; with an invertible transform the corresponding baseline in the explanation domain is then simple $\hat{z} = T(\hat{x})$. We will clarify this more explicitly in the revision.
>
> To further assess sensitivity to this choice, we repeated the Table 2 experiments using the dataset mean $\mathbb{E}[x]$ as an alternative neutral reference, shown below. We note that the "Avg." and "Zero" columns in Table 2 refer to the CPD substitution protocol rather than the IG baseline. This additional experiment reinforces two points. First, CDIG is not tied to a single hand-picked baseline: under the mean baseline, Cepstrum remains above TIMEX++ on all five datasets under both CPD protocols, while Fourier does so on 9/10 dataset/protocol pairs. Second, the choice of baseline materially affects the attribution values and can change the relative ordering between domains. We view this not as a weakness specific to CDIG, but as evidence that baseline choice is itself part of the interpretability setup, just like the choice of explanation domain.
>
> ||**PAM** || **Boiler** || **Epilepsy** || **Wafer** || **Freezer** ||
> |---|---|---|---|---|---|---|---|---|---|---|
> || Avg. | Zero | Avg. | Zero| Avg. | Zero| Avg. | Zero | Avg. | Zero |
> | Fourier | 0.047 | 0.436 | 0.782 | 0.995 | 0.14 | 0.143 | 0.34 | 0.34 | 0.273 | 0.273 |
> | Cepstrum | 0.386 | 0.371 | 1.317 | 0.39 | 0.226 | 0.709 | 0.826 | 0.812 | 0.33 | 0.431|
>
> More broadly, we view the baseline as part of the interpretability question being asked. A zero baseline asks which components of the observed signal contribute relative to absence of signal, whereas other baselines can support different questions. For example, in the PPG setting, using a previous-time-point sample as baseline would not measure contribution relative to "no signal", but rather contribution relative to a prior physiological state, attributing changes in heart rate or interference across time. We will revise the paper to make this distinction clearer and to state more explicitly that baseline choice should be aligned with the practitioner's explanatory goal.
>
> Q2. We view the gain as coming from both parts, but in different roles. The transform determines which semantic units become inspectable, while CDIG provides the attribution machinery in that chosen domain, with the same IG-style guarantees (e.g. completeness) across both real and complex transforms. This seperation is part of the contribution: it lets domain choice be studied under a fixed attribution principle, rather than coupling each representation to a different explanation rule.
>
> The experiments support this distinction. Holding the attribution backend fixed, different CDIG instantiations perform best on different datasets in Table 2, which shows that domain choice itself is consequential. Conversely, where overlap with prior methods exists (Table 3 in Appendix), CDIG remains directly competitive, which shows the contribution is not merely "choosing a good transform".
>
> Q3. We agree that direct empirical comparison is important wherever methods overlap. For that reason, Appendix F / Table 3 compares CDIG directly against FreqRISE and VIL-based baselines on AudioMNIST in the overlapping frequency and time-frequency settings. In those shared settings, CDIG is in the same faithfulness range as the specialised baselines, while achieving lower complexity than FreqRISE. At the same time, CDIG also extends to domains those baselines do not support, including the complex cepstrum.
>
> So our intent was not to argue only from generality. Rather, the evidence is twofold: direct competitiveness where prior cross-domain baselines exists, and broader applicability beyond those settings.
>
> Finally, we appreciate the point about over-interpretation. We will strengthen the limitations statement to make clear that CDIG is an attribution/faithfulness tool under a chosen baseline and explanation domain. It does not establish causal mechanisms, and clean-looking saliency maps can still be semantically misleading if the chosen domain is inappropriate. In high-stakes settings, we view these explanations as evidence to be triangulated with domain knowledge and intervention-based validation, not as definitive proof of model behaviour.

---

> > ### Author Rebuttal · Reviewer_XA1h · 2026-04-06
> >
> > Thank you for your response. I will leave my current score unchanged.

---

### Official Review · Reviewer_4xpx · 2026-03-13

**Soundness:** 3
**Presentation:** 3
**Significance:** 3
**Originality:** 3
**Overall Recommendation:** 4
**Confidence:** 3

**Summary:**

The work studies how to explain time series models in domains other than raw time, such as frequency, cepstrum, independent components, or trend/seasonality. The authors introduce Cross-Domain IG, which applies Integrated Gradients after mapping the input into a chosen transform domain. The paper presents the framework theoretically and then demonstrates it on several time series tasks, with examples from multiple applications.

**Compliance With Llm Reviewing Policy:**

Affirmed.

**Final Justification:**

The rebuttal has fully addressed my concerns. I will keep my recommendation score is 4.

**Key Questions For Authors:**

1. How should a practitioner choose the explanation domain in practice?
2. Can you say more about the computational overhead?

**Limitations:**

Yes.

**Strengths And Weaknesses:**

Strengths:
1. Soundness: the proposed method is a good idea. The paper argues that time-series saliency should not always be shown in the raw time domain, because important patterns may be easier to understand in other domains like frequency or trend or seasonality, which makes sense. The authors extends IG so explanations can be computed in a chosen transform domain.

2. Presentation: the paper is easy to follow.

3. Significance: in my opinion, the problem is important, especially for applications like healthcare or forecasting where raw time-point explanations may not be the most meaningful.

4. Originality: the work is interesting. The main novelty is turning transformed domain attribution into a general framework.



Weaknesses:
1. Soundness: my main concern is that the method depends a lot on picking the right domain. The examples are interesting, but the paper does not fully show when or how a user should choose the best transform in practice.

---

> ### Author Rebuttal · Authors · 2026-03-30
>
> We thank the reviewer for their thoughtful comments and for highlighting the importance of the practical domain-choice question.
>
> Q1. Our view is that the explanation domain should be chosen to answer the interpretability question, not treated as a free tuning knob. In practice, we follow a simple protocol that we will further clarify in the revision: (i) first state the signal one wants to inspect, (ii) then choose a transform whose coordinates correspond to that factor, and (iii) pre-specify that choice before looking at any saliency map.
>
> This is exactly the logic used in Section 5.1. We use Fourier when the question is about oscillatory/spectral structure, ICA when it is about source-level contributions, and STL when it is about trend vs. seasonality. The explanation domain changes across our case studies because the interpretability question changes across those applications.
>
> We do not claim that there is a single universally best transform for all time-series problems. In fact, one of the empirical takeaways of the paper is that different domain instantiations excel on different tasks/datasets. We therefore view domain choice as part of the interpretability specification: CDIG provides principled IG-style attributions once that specification is made. If multiple domains are plausible, our recommendation is to treat them as pre-specified complementary views and assess them with domain knowledge and faithfulness checks, rather than selecting one post hoc because it yields the most visually appealing map.
>
> We agree that this practical guidance should be stated more explicitly, and we will make this protocol clearer in the paper.
>
> Q2. CDIG is computationally close to standard IG. In both methods, the dominant cost comes from repeatedly evaluating gradients to approximate the path integral. The additional cost in CDIG comes from backpropagating through the inverse-transform step. As noted in Remark 4.6, CDIG preserves the same overall repeated-gradient computation pattern as IG with one extra transform-dependent step.
>
> For fixed transforms such as Fourier, this overhead is modest, since the dominant cost remains the repeated forward/backward passes already required by IG. For transforms such as ICA, there can also be an additional cost of constructing the representation. However, this cost is not specific to CDIG and would be incurred by any explainer operating in the ICA domain.
>
> We also ran additional controlled runtime experiments comparing standard IG and Fourier-domain CDIG. These show that CDIG adds a measurable but modest overhead, and that this overhead becomes a smaller fraction of total attribution time as model/input cost increases. In our tested settings, the overhead ranged from 27.68\% in a small shallow configuration to 1.77\% in the largest tested configuration. For completeness, we provide full runtime tables in our response to reviewer RFyj (Q3).
>
> We will clarify this computational tradeoff more explicitly in the revision.

---

> > ### Author Rebuttal · Reviewer_4xpx · 2026-04-03
> >
> > Thank you for your response. The rebuttal addressed my concerns. I am keeping my current score of 4.

---

### Decision · Program_Chairs · 2026-04-30

**Decision:**

Accept (spotlight)

**Comment:**

The authors build Cross-Domain Integrated Gradients. This method moves saliency maps from the time domain to any invertible and differentiable transform domain. It finds features in frequency or trend spaces that raw time points hide.

Reviewers find the work sound. Reviewer 4xpx praised the presentation and significance. Reviewer XA1h liked the logical approach and various application cases. Reviewer Kqv4 noted the thorough math and versatility. Reviewer RFyj called it a meaningful advancement with a useful open-source tool.